# Mitigating Social Desirability Bias in Random Silicon Sampling

## Abstract

Large Language Models (LLMs) are increasingly used to simulate population responses, a method known as "Silicon Sampling". However, responses to socially sensitive questions frequently exhibit Social Desirability Bias (SDB), diverging from real human data toward socially acceptable answers. Existing studies on social desirability bias in LLM-based sampling remain limited. In this work, we investigate whether minimal, psychologically grounded prompt wording can mitigate this bias and improve alignment between silicon and human samples. We conducted a study using data from the American National Election Study (ANES) on three LLMs from two model families: the open-source Llama-3.1 series and GPT-4.1-mini. We first replicate a baseline silicon sampling study, confirming the persistent Social Desirability Bias. We then test four prompt-based mitigation methods: *reformulated* (neutral, third-person phrasing), *reverse-coded* (semantic inversion), and two meta-instructions, *priming* and *preamble*, respectively encouraging analytics and sincerity. Alignment with ANES is evaluated using Jensen-Shannon Divergence with bootstrap confidence intervals. Our results demonstrate that reformulated prompts most effectively improve alignment by reducing distribution concentration on socially acceptable answers and achieving distributions closer to ANES. Reverse-coding produced mixed results across eligible items, while the Priming and Preamble encouraged response uniformity and showed no systematic benefit for bias mitigation. Our findings validate the efficacy of prompt-based framing controls in mitigating inherent Social Desirability Bias in LLMs, providing a practical path toward more representative silicon samples.

## 1 Introduction

Large Language Models (LLMs) Radford et al. (2018); Kojima et al. (2022) can simulate human emotions and opinions, from subjective labeling of Twitter posts (Törnberg, 2023; Yang et al., 2024) and participation in psychological studies (Aher et al., 2023; Qiu & Lan, 2024) to producing behavior changes consistent with personality frameworks (Serapio-García et al., 2023; Besta et al., 2025). These non-trivial capabilities naturally led researchers to explore whether LLMs can be used to simulate entire populations for social sciences Argyle et al. (2023); Yang et al. (2024); Gao et al. (2023), polling Yu et al. (2024); Zhang et al. (2024), or marketing research studies Sarstedt et al. (2024); Arora et al. (2025). Using LLM-simulated respondents in these setups could help tackle the limitations of large sample sizes, high costs, and long execution times associated with human respondents. This led to the idea of *silicon sampling*, which refers to the use of LLM-generated agents with demographic conditioning to simulate population-level survey responses Argyle et al. (2023); Sun et al. (2024).

Prior work found remarkable alignment between silicon and human samples on some topics; however they diverged more sharply when sensitive topics or groups were involved Sun et al. (2024). This divergence likely reflects *Social Desirability Bias (SDB)*, i.e., the tendency of LLMs to generate socially approved rather than demographically representative answers (Salecha et al., 2024) (see Section 2).

To make silicon sampling a viable method, reliability across topics is of key importance, which makes investigating this divergence an important research direction. We believe that the social desirability bias

largely stems from the otherwise benign decision of model developers to suppress humans' harmful biases and stereotypes learned by the model, a finding that is repeatedly confirmed by research that finds almost no explicit stereotypes in responses of different models Liang et al. (2021); Lin et al. (2024); Bai et al. (2025); Zhao et al. (2025); Li et al. (2025). When performing silicon sampling, however, it is desirable to elicit a model's knowledge about stereotypes held by different groups of people to yield more accurate results. Although models have been successfully trained to avoid displaying any *explicit* stereotypes, researchers still find substantial *implicit* stereotypes when models are queried in a more indirect manner that does not provoke "defensive" behavior Bai et al. (2025); Zhao et al. (2025). This suggests that careful prompt design White (2023) may reduce the social desirability bias and achieve more representative responses. In a silicon sampling setting, relying entirely on implicit querying methods is not feasible. However, we hypothesize that it is possible to reduce the social desirability bias by using other methods, drawn from previous LLM and psychological research.

The main goal of this study lies in the systematic exploration of prompt engineering techniques for reducing the social desirability bias in large population silicon sampling Sun et al. (2024). We test four prompt design strategies to evaluate their effectiveness in mitigating bias and bringing a model's responses closer to human populations. This way, we hope to provide future silicon sampling studies with a method of achieving closer alignment with human samples, particularly on sensitive topics. The main contributions of this work are:

- We present the first systematic evaluation of prompt-based mitigation strategies for social desirability bias in large-scale silicon sampling with LLMs.
- We establish a controlled experimental benchmark by replicating prior silicon-sampling results and extending them with four prompt manipulation conditions, enabling a comparative analysis against human survey distributions.
- We demonstrate that question reformulation, neutralizing wording and adopting third-person framing, generally improves distributional alignment and response diversity, while other strategies (e.g., reverse-coding, priming, and preamble) yield limited or inconsistent benefits.
- We further show that higher-stochasticity decoding minimally improves alignment across all prompt conditions, and question reformulation still possesses the relative advantage among others.
- To ensure robustness of our findings, we control for the impact of variation in populations on our results. We demonstrate the robustness of our conclusions under demographic stratification within survey as well as between survey waves (ANES 2020 vs. 2024).

## 2 Related Work

**Silicon sampling and synthetic populations** Researchers have examined population-level simulations using LLMs, an approach known as *silicon sampling* (Argyle et al., 2023; Lee et al., 2024; Sarstedt et al., 2024). Lee et al. (2024) investigated the algorithmic fidelity and biases of LLMs in simulating public opinions regarding global warming. Instead, Sun et al. (2024) used demographic distributions from the American National Election Study (ANES) to construct synthetic respondents and compared their political survey responses with those of real participants. While there was significant alignment between silicon and human samples on objective or politically neutral items, a larger disagreement was observed for socially sensitive questions, especially about racial diversity, gender equality, or identity politics Sun et al. (2024). The results indicate that LLMs may tend to produce responses that align with socially desirable or politically neutral positions rather than the human opinions in real populations.

**Bias and social desirability in LLMs** There is a broader body of research on bias in modern LLMs Schramowski et al. (2022); Gallegos et al. (2024) that provides insight on diverging answers in silicon sampling when dealing with sensitive subjects. Many LLMs undergo the process of alignment training, fine-tuning language models to follow human values and avoid harmful or biased outputs Bai et al. (2025); Zhao et al. (2025); Li et al. (2025). While this process reduces explicit bias, implicit biases can remain embedded in internal representations and emerge only under indirect or carefully crafted queries. This aligns with evidence that LLMs often exhibit social desirability bias, producing socially approved rather than statistically representative answers Salecha et al. (2024). It is further revealed that the extent of social desirability bias depends on the context. In particular, models show stronger social desirability bias in the context of

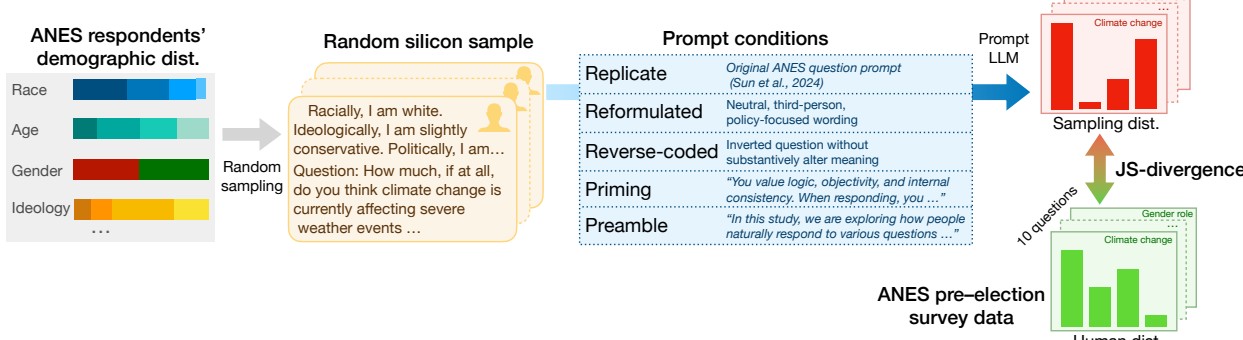

Figure 1: Overview of the experimental pipeline.

being "judged", mirroring findings from human psychology, where respondents offer more socially desirable answers when under the impression of being evaluated Crowne & Marlowe (1960); Paulhus & Reid (1991).

**Mitigation strategies and cognitive framing**  Research in Psychology and Computer Science suggests approaches to reduce social desirability bias through changes in framing and context. In human respondents, bias can be mitigated by rephrasing evaluative questions into neutral or third-person formats, and by assuring anonymity and non-judgment Fisher (1993). Similarly, in LLMs, prompt conditioning and framing can influence the response mode of the model Park et al. (2023a;b); Besta et al. (2025). For example, by pre-conditioning models to adopt a more analytical persona, researchers have been able to produce more consistency in models' strategies in a game setting, with less adaptation to social circumstances. The results were also more representative of the real-world answers and less constrained by social expectations Besta et al. (2025).

**Summary and research gap**  Prior work shows that LLMs can simulate human-like personas and that silicon sampling can approximate population-level survey responses. However, responses to socially sensitive questions often diverge from empirical data due to social desirability bias. Although alignment training helps reduce explicit bias, it may also suppress implicit biases that contribute to realistic human behavior in simulations. Despite this, little is known about how prompt framing, evaluation context, or cognitive conditioning influences SDB in LLM-based population sampling. We address this gap by systematically evaluating psychologically grounded prompt interventions, such as neutral phrasing, third-person framing, and rational-mode prompting, and their effects on alignment between silicon and human responses.

## 3  Methodology

Our methodology follows a four–stage pipeline: (1) extracting demographic distributions from the ANES 2020 dataset; (2) generating a synthetic (silicon) population by conditioning LLMs on these demographic profiles; (3) collecting survey responses under five prompt conditions; and (4) evaluating alignment between silicon and human responses using divergence-based metrics. The experimental pipeline is illustrated in Figure 1. Data, code, and results are at `https://anonymous.4open.science/r/mitigate-social-desirability-bias-B092/`.

### 3.1  Data and sampling

We mainly use the American National Election Studies (ANES) 2020 pre-election survey dataset, which includes 5,441 respondents.[1] The dataset contains both demographic information and responses to a wide range of political and social questions. We focus on ten multiple–choice questions covering different social and political topics, including *Racial Diversity*, *Gender Role*, *Current Economy*, *Drug Addiction*, *Climate Change*, *Gay Marriage*, *Refugee Allowing*, *Health Insurance*, *Gun Regulation*, and *Income Inequality*.

---

[1]`https://electionstudies.org/data-center/2020-time-series-study/`

To generate synthetic respondents, we follow the random silicon sampling procedure of Sun et al. (2024). Let $\mathcal{D}_{\text{ANES}}$ denote the empirical ANES population. From $\mathcal{D}_{\text{ANES}}$, we estimate empirical marginal distributions over $K = 8$ demographic variables $\{D_1, D_2, \ldots, D_K\}$, corresponding to *race*, *age*, *gender*, *ideology*, *party identification*, *church attendance*, *political interest*, and *political discussion frequency*. We then construct a silicon sample $\{R_i\}_{i=1}^N$ with $N = 5{,}441$, matching the size of the human dataset. Each synthetic respondent $R_i$ is defined by a demographic profile sampled independently from these empirical marginals:

$$R_i = \{d_k^{(i)} \sim D_k\}_{k=1}^K.$$

The demographic attributes of each $R_i$ are rendered in natural language and provided to the language model as conditioning context, together with a survey question. The language model then generates a response for that synthetic individual. Repeating this procedure across all $R_i$ yields an aggregate response distribution from random silicon subjects. Details of demographic variables are in Appendix A.9.

### 3.2 Model selection

Our analysis employs three LLMs to compare performance across scale and licensing: the open-source Llama-3.1-8B-Instruct (Llama-8B) and Llama-3.1-70B-Instruct (Llama-70B) AI (2024), and the closed-source GPT-4.1-mini (released on 2025-04-14) Achiam et al. (2023). The closed-source model represents an advancement over the model (i.e., GPT-3.5-turbo released on 2023-06-13) used in prior silicon studies Sun et al. (2024). We selected GPT-4.1-mini over the GPT-5 family (released on 2025-08-07) for its optimal performance-cost balance and faster inference speed, making it better suited for high-throughput survey simulations. The selected models allow us to evaluate the trade-off between model scales, capability, computational cost, and reproducibility. To ensure consistent comparison and maximal reproducibility, all models were configured for deterministic output. More implementation details are provided in Appendix A.1.

### 3.3 Prompt-based mitigation strategies

The core of silicon sampling involves generating responses by conditioning the LLM on unique demographic profiles derived from the ANES dataset. To mitigate Social Desirability Bias (SDB), we design five experimental conditions by systematically varying the question structure, language, and contextual instructions.

**Replicate Condition (0)** We replicate the results of prior work Sun et al. (2024). We keep the prompts unchanged, and ask questions directly copied from the ANES dataset. This condition is used as a baseline to compare other conditions, as well as a measure of comparative performance of three LLMs on this task.

**Reformulated Condition (1)** We attempt to minimize SDB by reducing the perception of being evaluated or asked for an explicit opinion as determined by the LLM. This follows prior work showing that LLMs tend to produce more socially desirable answers, particularly when evaluated, and that alignment training controls explicit but not implicit bias Salecha et al. (2024); Bai et al. (2025). We apply several modifications to the phrasing of the survey questions while ensuring that their meaning remains unchanged:

- Following psychological research, we neutralize the questions by using less evaluative language (e.g., avoiding evaluative words like "good", "positive", "bad").
- We re-formulate the questions to avoid direct "what do you think" phrasing, using a more neutral third-person formulation (e.g., "what would this respondent think").
- Where possible, questions are re-formulated to ask about one's opinion about a policy rather than a social phenomenon or a group.

An example of a typical prompt under Reformulated Condition:

> *Racially, the respondent is black. The respondent doesn't like to discuss politics with my family and friends. Ideologically, the respondent is slightly liberal. Politically, the respondent is an independent. The respondent does not attend church. The respondent is 43 years old. The respondent is a man. The respondent is somewhat interested in politics.*

*How would this respondent assess if there should be an increase, decrease, or no change in government spending to help people pay for health insurance when people cannot pay for it all themselves?*

*1. Increase 2. Decrease 3. No change*

**Reverse-coded Condition (2)**    Prior work found that reverse-coding of questions can reduce SDB Salecha et al. (2024). To test whether it can alleviate social desirability on sensitive topics, we include reverse-coded versions *where this does not substantively alter the semantic meaning of the item.* An example is given below:

Original: How much, if at all, do you think climate change is currently affecting severe weather events or temperature patterns in the United States?

*Reverse-coded: How much do you agree with the statement that climate change is unrelated to severe weather events or temperature patterns in the United States?*

For reversed-coded questions, the resulting answer options were inverted numerically to ensure consistency in the final distribution analysis. Items where reverse-coding alters the construct are omitted.

**Priming Condition (3)**    This condition is grounded in studies showing that psychological priming of LLMs can influence behaviour Besta et al. (2025). We use the conditioning prompts to create a more "Thinking" agent, added before the survey question. The verbatim instructions are:

You value logic, objectivity, and internal consistency. When responding, you prioritize reasoning over emotion, and aim to base your answers on evidence, structure, and rational analysis. Your goal is to provide clear, well-reasoned, and intellectually honest answers that reflect careful thought. The best answer is that which reflects your reasoning process.

**Preamble Condition (4)** Adding a preamble encouraging sincere answers and promising no judgment is standard in studies on sensitive topics as it reduces pressure to conform among human respondents. In the LLM setting, it may conversely trigger a perception of evaluation and increase conformity; we therefore include a preamble condition before the question to test this. We use the following verbatim preamble:

In this study, we are exploring how people naturally respond to various questions. To ensure meaningful results, it is important that responses reflect your genuine thoughts and feelings. There are no correct or desirable answers, and your responses will not be evaluated or judged. Please answer honestly and without concern for how your answers might be perceived. Your sincerity helps us better understand authentic human responses.

For each condition, the same distribution of demographic variables was used, and each new response was produced with a *fresh and isolated LLM session* to preclude any memory or carry-over of context. To standardize data collection, all user prompts listed the discrete answer options and explicitly instructed the model to *respond with a single number only.* All questions and prompt conditions, including reformulated and reverse-coded variants are in Appendix A.8. LLM prompting examples are in Appendix A.10.

## 3.4   Evaluation

**Jensen-Shannon divergence (JS-divergence)**    To quantify alignment between LLM-generated ("silicon") and human survey responses, we use JS-divergence. For each multiple-choice question, we compute the JS-divergence between the empirical human response distribution from ANES and the corresponding silicon response distribution under each experimental condition. JS-divergence is a symmetric, bounded measure of distributional similarity ($[0, 1]$), making it well-suited for comparing response similarity. It is derived from the Kullback-Leibler Divergence (KL-divergence), defined as:

$$D_{JS}(P\|Q) = \frac{1}{2}D_{KL}(P\|M) + \frac{1}{2}D_{KL}(Q\|M), \tag{1}$$

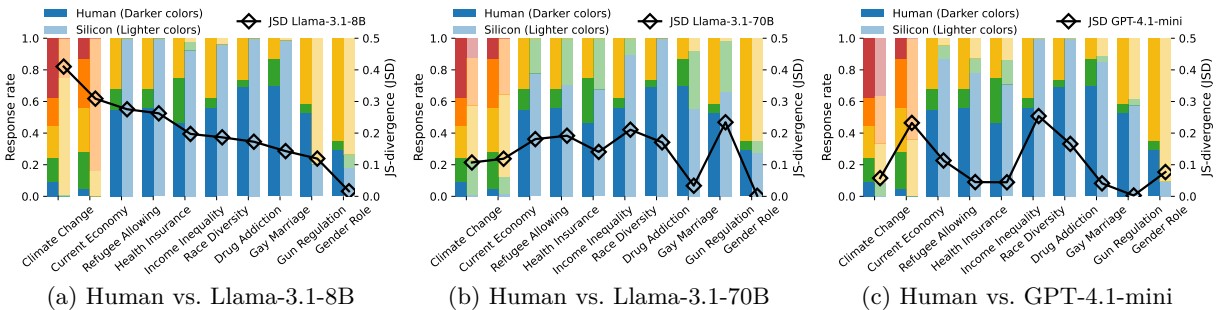

(a) Human vs. Llama-3.1-8B  (b) Human vs. Llama-3.1-70B  (c) Human vs. GPT-4.1-mini

Figure 2: Baseline replication results for ten questions. Darker/lighter colors indicate answers of ANES/simulated respondents. Black line shows the JS divergence score based on the replicate results.

$$D_{KL}(P\|M) = \sum_{x \in X} P(x) \log \frac{P(x)}{M(x)}, \tag{2}$$

where $P$ and $Q$ represent the human and silicon response distributions over answer options $x \in X$ for a given question. $M = \frac{1}{2}(P + Q)$ is a mixture distribution of $P$ and $Q$. Lower JS-divergence values indicate closer alignment between the human and silicon samples.

**Experimental conditions** For each question $X$, we evaluate five experimental conditions $c \in \{0, 1, 2, 3, 4\}$. Let $P_X$ denote the human response distribution for question $X$, and $Q_{c,X}$ the silicon response distribution under condition $c$. This yields five JS-divergence values per question:

$$D_{JS}(X, c) = D_{JS}(P_X\|Q_{c,X}).$$

**Uncertainty estimation and statistical comparison** To estimate uncertainty and assess whether differences between conditions are statistically meaningful, we apply a non-parametric bootstrap over silicon responses. For each question $X$ and condition $c$:

1. We sample with replacement from the silicon responses to obtain bootstrap replicates $Q_{c,X}^{(j)}$, where $j = 1, \ldots, n$.
2. For each replicate $j$, we compute the corresponding JS-divergence $D_{JS}^{(j)}(X, c)$, yielding a bootstrap distribution of divergence values.

We then compute 95% confidence intervals from the empirical percentiles of each bootstrap distribution. Comparing these intervals across conditions allows us to evaluate which silicon response distributions are more statistically aligned with human data. All bootstrap analyses are performed with $n = 2,000$.

## 4 Experimental Results

### 4.1 Results across experimental conditions

**Replicate Condition (0)** We replicated the base silicon-sampling pipeline Sun et al. (2024) using our selected LLMs, conditioning each sample on standard demographic attributes and the original survey question. Figure 2 compares the human (ANES 2020) and LLM-simulated response distributions for ten multiple–choice questions. The smallest open-source model, Llama-8B, exhibits substantial drift from the human distributions on most questions. In particular, it frequently collapses to a single, socially favorable option on sensitive items, as evidenced by dominant lighter-blue bars for questions such as *Refugee Allowing*, *Health Insurance*, *Drug Addiction*, and *Gay Marriage*. This behavior reflects strong mode collapse and limited response diversity, possibly characteristic of social desirability bias (SDB) in small-capacity models.

The larger Llama-70B shows greater response diversity and generally lower JS-divergence than Llama-8B. However, a counterintuitive pattern emerges: on several questions (e.g., *Refugee Allowing*, *Health Insurance*, and *Income Inequality*), the model disproportionately selects both the most socially desirable (blue)

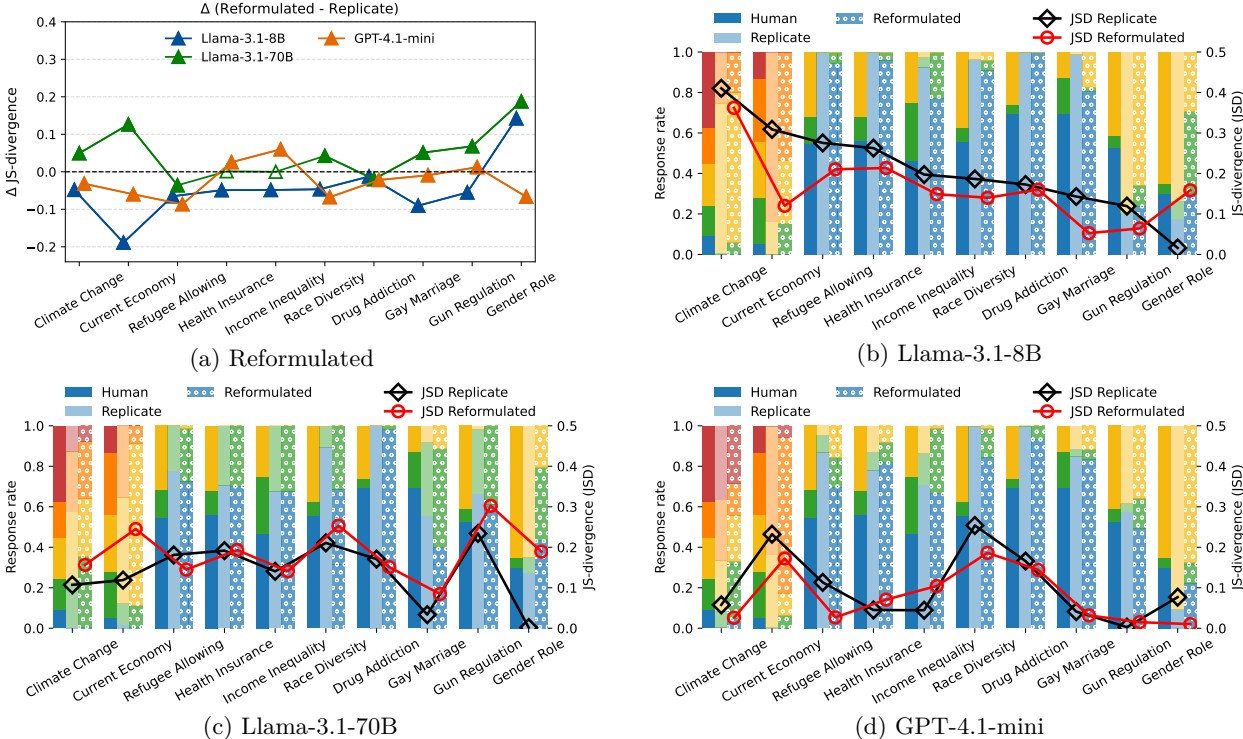

(a) Reformulated

(b) Llama-3.1-8B

(c) Llama-3.1-70B

(d) GPT-4.1-mini

Figure 3: (a) Difference in JS-divergence ($\Delta$) between Reformulated and Replicate conditions. Negative values indicate an improvement (closer alignment to ANES) achieved by the condition. Solid triangles indicate statistical significance at the 95% confidence level. (b-d) Human, Replicate, and Reformulated responses for ten questions on three LLMs.

and the socially undesirable (green) options. In contrast, human respondents more frequently choose neutral/moderate responses (yellow), such as "No change" or "Neither favor nor oppose". This suggests that while increased model capacity alleviates mode collapse, it may also amplify latent ideological or training-induced biases, leading the model to favor polarized responses.

For GPT-4.1-mini, we observe a different failure mode. Across most questions, the model over-selects socially acceptable (blue) or moderate (yellow) options, resulting in narrower and more uniformly positive distributions than those observed in ANES. On particularly sensitive questions such as *Race Diversity* and *Drug Addiction*, the model produces nearly identical responses for all simulated individuals, while human answers to the same question were a lot more diverse.

Overall, JS-divergence scores indicate that GPT-4.1-mini produces distributions statistically closer to the ANES data than Llama models, although performance varies by question (e.g., *Race Diversity* vs. *Gender Role*). These observations confirm the presence of SDB in the baseline condition, manifesting as mode collapse or over-moderation, and motivate the need for SDB mitigation strategies.

**Reformulated Condition (1)** In this condition, questions are neutralized and expressed in a third-person perspective. Figure 3a reports the difference in JS-divergence between the Reformulated and Replicate conditions, i.e., $\Delta = D_{JS}(X, 1) - D_{JS}(X, 0)$, where negative values indicate improved alignment with the ANES data. Both Llama-8B and GPT-4.1-mini exhibit clear improvements under reformulation, with negative $\Delta$ values for most questions, many of which are statistically significant. Specifically, reformulation improves alignment on 9 out of 10 questions for Llama-8B and on 6 out of 10 questions for GPT-4.1-mini.

Beyond reduced JS-divergence, the Reformulated condition consistently yields greater response diversity. As shown in Figure 3b (Llama-8B) and Figure 3d (GPT-4.1-mini), a larger proportion of generated respondents (bars with hatch "oo") select less socially desirable options, resulting in response distributions that more

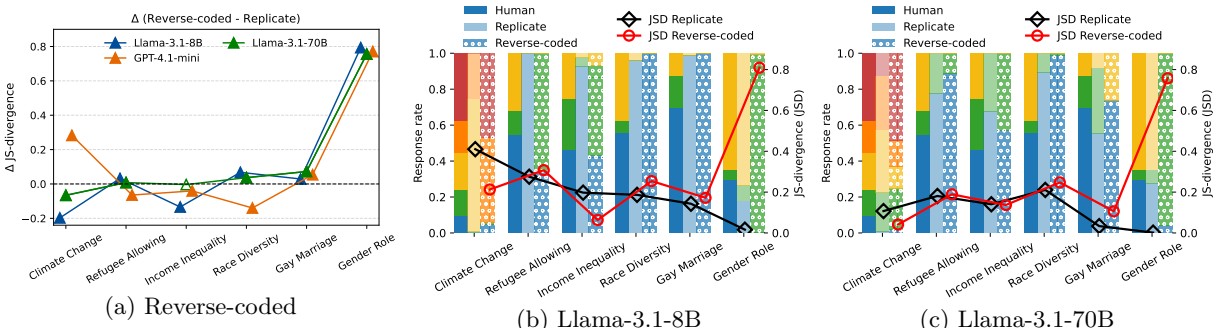

Figure 4: (a) Difference in JS-divergence (Δ) between Reverse-coded and Replicate conditions. Negative values indicate an improvement (closer alignment to ANES) achieved by the condition. Solid triangles indicate statistical significance at the 95% confidence level. (b-c) Human, Replicate, and Reverse-coded responses for six questions on Llama models.

closely resemble human data. For example, for *Race Diversity*, a substantial share of responses shifts from the most socially acceptable option ("Better", blue) toward the more controversial option ("Worse", green). Similar patterns exist for *Refugee Allowing* and *Gender Role*. Notably, *Gender Role* is the single item where reformulation worsens performance for Llama-8B. However, we still observe an increased diversity pattern, where responses are less concentrated on the safe "Makes no difference" option (yellow) and spread across more alternatives. Human responses to this question are relatively non-controversial (more on the safe option in yellow), likely reflecting that household gender roles are comparatively settled in contemporary U.S. society. This highlights a key limitation: mitigating SDB is insufficient if the model's underlying beliefs about a demographic group are inaccurate. When an LLM's internal representations diverge from empirical reality, reducing SDB can push the simulated distribution further from human data.

Llama-70B shows limited gains from reformulation, with significant improvements on only 2 out of 10 questions based on JS-divergence differences. Nevertheless, closer inspection of the simulated responses reveals notable qualitative changes. In Figure 3c, reformulation increases response diversity and shifts probability mass toward less socially desirable options, i.e., more "Oppose" (green) and "Neither favor nor oppose" (yellow), and fewer safe "Favor" (blue) responses, for *Race Diversity* and *Gay Marriage*. The same pattern is observed for *Gender Role*, where the responses shift from "Makes no difference" (yellow) to "Worse" (green). Despite an increased proportion of less socially acceptable responses, the resulting distributions turn out misaligned with human data, leading to higher JS-divergence overall. This might be explained by the higher polarization of responses in this model's Replicate condition. Starting with an already more "controversial" baseline, further reduction in SDB moved the distribution farther from the more moderate human responses.

Overall, these results provide consistent evidence that question reformulation can mitigate the impact of SDB by increasing response diversity and reducing over-selection of socially safe answers.

**Reverse-coded Condition (2)** The Reverse-coded condition applies to only six questions for which reverse-coding was deemed feasible. The results are unstable and its effectiveness as a SDB mitigation strategy is unclear. Figure 4a shows that both Llama models exhibit improvements on *Climate Change*, with Figure 4b and Figure 4c indicating the response distributions more closely resemble the human data. However, this conclusion does not apply to other questions or to GPT-4.1-mini.

A consistent pattern of degradation appears for *Gender Role*, where all models show substantial worsening. This stems from the difficulty of preserving semantic equivalence when implementing reverse-coding. The original question evaluates attitudes toward a situation, "*the man works outside the home and the woman takes care of the home and family.*" In this condition, it was modified to "*both the man and the woman share work outside the home and take care of the home and family.*" We consider this the closest plausible reverse-coded variants, yet it arguably alters the underlying construct rather than merely reversing its polarity. Such semantic shifts likely account for the pronounced misalignment observed for this question.

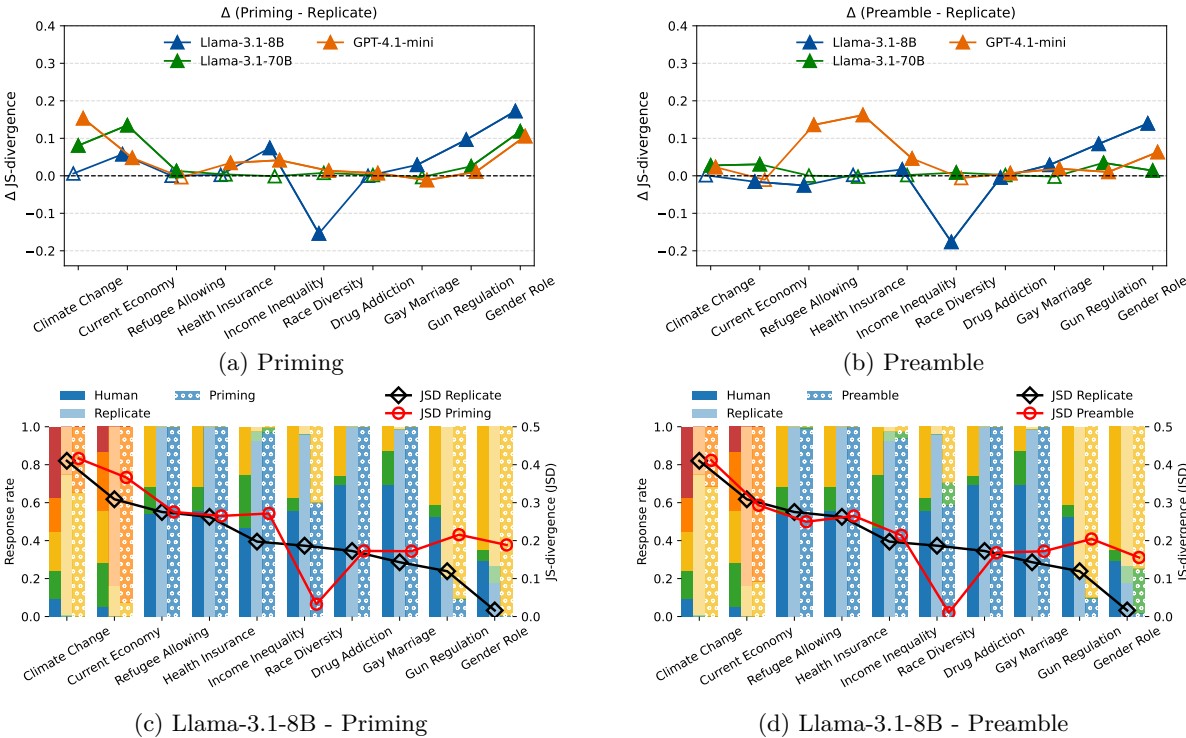

(a) Priming

(b) Preamble

(c) Llama-3.1-8B - Priming

(d) Llama-3.1-8B - Preamble

Figure 5: Difference in JS-divergence ($\Delta$) between Priming, Preamble conditions, and Replicate conditions. Negative values indicate an improvement (closer alignment to ANES) achieved by the condition. Solid triangles indicate statistical significance at the 95% confidence level. (c-d) Human, Replicate, and Priming/Preamble responses for ten questions on Llama-8B.

**Priming (3) and Preamble (4) Conditions** Figure 5a and Figure 5b illustrate the change in JS-divergence for the Priming and Preamble relative to the Replicate condition. Overall, adding either a rational primer or a preamble encouraging sincere responses does not mitigate SDB. While we observe reductions in JS-divergence for Llama-8B on *Race Diversity*, accompanied by increased response diversity and greater overlap with human data (see Figure 5c and Figure 5d), these gains are inconsistent and often reversed on other questions. For Llama-70B and GPT-4.1-mini, both Priming and Preamble increase JS-divergence, and lower response diversity in many cases compared to the Replicate condition (see Appendix Figure 7).

The degradation in Priming and Preamble is often driven by increased response uniformity, as model outputs concentrate more heavily on socially acceptable options. We hypothesize that explicitly instructing models to be "truthful" or "sincere" may inadvertently activate a perception of evaluation in the model, reinforcing socially safe patterns, as LLMs notice they are being monitored and adjust their behavior accordingly. This observation aligns with a recent study on LLMs' agentic behavior where models made more conforming choices under evaluation compared to when they perceived the task as real deployment Lynch et al. (2025). Rather than reducing SDB, such conditions appear to amplify it, resulting in less representative distributions.

## 4.2 Effect of decoding stochasticity

Beyond prompt design, decoding strategy may influence SDB outcomes depending on the level of stochasticity employed during sampling. We therefore examine the effect of decoding temperature on response alignment. Table 1 reports the average JS-divergence for GPT-4.1-mini at two decoding temperatures. Increasing the temperature from 0 to 1 reduces JS-divergence in the Replicate condition, indicating that a portion of SDB arises from low-entropy decoding (low stochasticity). Higher-temperature decoding allows the model to sample from a broader set of plausible responses, mitigating model collapse toward socially acceptable options. Yet, the improvement is modest, indicating that decoding strategy alone cannot fully address SDB.

Table 1: Average JS-divergence for GPT-4.1-mini across ten questions under different prompting strategies and decoding temperatures. Lower values indicate better alignment with human responses.

| | Replicate | Reformulated | Reverse-coded | Priming | Preamble |
|---|---|---|---|---|---|
| T = 0 | 0.1033 | 0.0787 | 0.2415 | 0.1429 | 0.1483 |
| T = 1 | 0.0901 | **0.0678** | 0.2302 | 0.1347 | 0.1274 |
| % ↓ (T=0→1) | 12.8% | 13.9% | 4.7% | 5.7% | 14.1% |

Reformulated condition consistently outperforms all other prompting strategies at both temperatures, achieving the lowest JS-divergence at T=1 (0.0678). These results indicate that reformulation improves alignment with human response distributions in random silicon sampling and further benefits from the increased sampling diversity induced by stochastic decoding.

Reverse-coded prompting still performs poorly on average under stochastic decoding (T=1), with only marginal improvement (4.7%). Higher-entropy decoding appears insufficient to resolve human–silicon alignment issues, as simply increasing randomness cannot overcome the semantic instability inherent in reverse wording. Priming and Preamble also remain misaligned with human responses at higher decoding temperatures (T=1). Though both benefit modestly from increased stochasticity (5.7% and 14.1%), their JS-divergence remains higher than the Replicate and Reformulated baselines. It suggests that the biases introduced by explicit instructional framing are structural and not mitigated by increased sampling entropy.

### 4.3 Demographic-stratified results

Table 2 summarizes how the four mitigation strategies, Reformulated, Reverse-coded, Priming, and Preamble, affect simulation alignment across seven *aggregated* demographic groups using GPT-4.1-mini.[2] We focus on three representative questions that cover sensitive sociopolitical topics: *Race Diversity* that exhibits consistent SDB effects, and *Refugee Allowing* and *Income Inequality* which show mixed SDB effects across LLMs in the Replicate/baseline setting.

Reformulated condition consistently reduces JS-divergence for all seven groups on the social and political topics, *Race Diversity* and *Refugee Allowing*. This is potentially because the neutral, third-person paraphrasing helps the model ignore the considerable social pressure related to the topics, resulting in more nuanced distributions. The condition fails on the economic issue *Income Inequality*. This is likely caused by the relatively lower political sensitivity of economic questions compared to, for example, issues of race and gender. In this case, the reductions in SDB provide limited improvements, while question reformulation may introduce slight semantic shifts, resulting in responses further away from the human distribution. Reverse-coded condition achieves the greatest overall improvements in these specific questions, substantially lowering divergence for all groups on *Race Diversity* (e.g., −0.151 to −0.193) and meaningfully improving alignment on *Refugee Allowing* and *Income Inequality* for most groups. Though promising, these effects do not generalize to other eligible questions or models (see Appendix Table 13, Table 17, Table 21).

Priming and Preamble exhibit consistent patterns between the subgroups similar to the Reformulated condition, despite frequently producing less alignment with the human distribution. This implies that subgroup sensitivity is not driven by any *single demographic axis* but instead reflects broader structural properties of how LLMs emulate population-conditioned responses. Analysis of fine-grained subgroups (e.g., White vs. Black vs. Asian) follows similar trends but reveals additional nuance. Full results are in Appendix A.7.

### 4.4 Robustness to temporal and demographic shift

A key question for silicon sampling is whether prompt-based mitigation strategies remain effective under temporal shifts in survey populations and response patterns. To evaluate this, we compare the Reformulated condition on eight overlapping questions between ANES 2020 and the newly released ANES 2024 survey.[3].

---

[2]We exclude *age* from the stratified analysis because it is a continuous variable that requires arbitrary discretization into ranges, which could introduce confounding design choices and obscure subgroup effects.

[3]https://electionstudies.org/data-center/2024-time-series-study/

Table 2: Difference in JS-divergence ($\Delta$) between four prompt conditions and Replicate on stratified groups using GPT-4.1-mini. Darker green signifies greater improvement in alignment ($\Delta < 0$), while darker red signifies greater worsening of alignment ($\Delta > 0$).

| Group | Reformulated | | | Reverse-coded | | | Priming | | | Preamble | | |
|---|---|---|---|---|---|---|---|---|---|---|---|---|
| | Race Diversity | Refugee Allowing | Income Inequality | Race Diversity | Refugee Allowing | Income Inequality | Race Diversity | Refugee Allowing | Income Inequality | Race Diversity | Refugee Allowing | Income Inequality |
| Race | -0.05 | -0.112 | 0.065 | -0.161 | -0.096 | -0.044 | 0.004 | -0.016 | 0.026 | -0.02 | 0.114 | 0.023 |
| Discuss Politics | -0.105 | -0.142 | 0.067 | -0.16 | -0.083 | -0.06 | 0.025 | 0.017 | 0.041 | -0.011 | 0.153 | 0.059 |
| Ideology | -0.059 | -0.071 | 0.05 | -0.129 | -0.063 | 0.108 | 0.019 | 0.009 | 0.076 | -0.008 | 0.125 | 0.067 |
| Party | -0.07 | -0.083 | 0.04 | -0.134 | -0.061 | 0.065 | 0.013 | 0.015 | 0.045 | -0.007 | 0.125 | 0.061 |
| Church | -0.068 | -0.087 | 0.062 | -0.137 | -0.061 | -0.038 | 0.012 | -0.002 | 0.044 | -0.007 | 0.138 | 0.047 |
| Gender | -0.069 | -0.085 | 0.06 | -0.14 | -0.064 | -0.037 | 0.013 | -0.003 | 0.042 | -0.007 | 0.137 | 0.047 |
| Political Interest | -0.083 | -0.08 | 0.043 | -0.159 | -0.074 | -0.048 | 0.019 | 0.004 | 0.036 | -0.003 | 0.155 | 0.054 |

Table 3: Difference in JS-divergence between Reformulated and Replicate conditions for overlapping questions in ANES 2020 and 2024, using GPT-4.1-mini. Negative values indicate improved alignment and * denotes statistical significance.

| | | Climate Change | Health Insurance | Income Inequality | Race Diversity | Drug Addiction | Gay Marriage | Gun Regulation | Gender Role |
|---|---|---|---|---|---|---|---|---|---|
| 2020 | Replicate | 0.058 | 0.045 | 0.044 | 0.254 | 0.166 | 0.041 | 0.003 | 0.077 |
| | Reformulated - Replicate | -0.032* | 0.026* | 0.060* | -0.068* | **-0.022**\* | -0.009* | 0.012* | **-0.067**\* |
| 2024 | Replicate | 0.086 | 0.029 | 0.023 | 0.293 | 0.123 | 0.032 | 0.001 | 0.028 |
| | Reformulated - Replicate | **-0.050**\* | 0.004 | 0.087* | **-0.110**\* | -0.005* | **-0.026**\* | 0.016* | -0.012* |

As shown in Table 3, reformulation produces reductions in JS-divergence for many questions in ANES 2024, with particularly pronounced gains for *Climate Change*, *Race Diversity*, and *Gay Marriage* compared to 2020. The overall consistency of these improvements across ANES 2020 and 2024 suggests that reformulation is a general mechanism for mitigating social desirability bias in LLM-generated survey responses, not tied to a specific survey snapshot.

We further examined demographic shifts between ANES 2020 and 2024. Compared to 2020, the 2024 data show slightly more conservative respondents, greater polarization in self-reported party identification, and marginally lower political interest on average. In addition, non-informative responses for church attendance increased from 0.8% to 5.1%, mainly due to the introduction of an "Inapplicable" option in the 2024 survey. More details are in Appendix A.4. Although these changes are not large enough to substantially alter overall response distributions, they provide a meaningful robustness test. Under these shifts, neutral, third-person question framing remains a promising approach for reducing divergence from human responses.

## 5 Conclusion

This work presents a systematic study of prompt-level mitigation strategies for Social Desirability Bias (SDB) in LLM-based survey simulation. Building on random silicon sampling, we formalize a set of mitigation strategies, including neutral reformulation, reverse-coding, priming, preamble, and evaluate them under a unified experimental framework. Using ANES data, we quantify their effects across models, questions, demographic strata, decoding stochasticity, and survey waves, enabling the first controlled comparison of SDB mitigation mechanisms in LLM-generated survey responses.

Our results identify question reformulation as the most effective and consistent mitigation strategy. Neutral, third-person rephrasing reduces evaluative pressure in question wording, consistently leading to *more diverse response distributions*. In most cases, this also leads to substantially lower divergence from human data. In a few cases where the baseline responses were already "controversial", reformulation shifted the distribution further away from the more moderate human distribution. The observations also hold across survey years with different demographic distributions.

Reformulation does not fully eliminate SDB. Politically sensitive or culturally entrenched topics continue to elicit concentrated responses, particularly for smaller models, indicating that mitigation is constrained by model capacity and underlying population knowledge. Other strategies show limited or inconsistent

benefits. Reverse-coding is highly sensitive to semantic fidelity. Priming and preamble often increase response uniformity, suggesting that simple instructions or framing are insufficient to resolve SDB and may even exacerbate it.

Demographic-stratified analyses show that SDB is largely structural rather than driven by idiosyncratic effects tied to specific subgroups. LLM temperature analysis further demonstrates that increased decoding stochasticity alleviates mode collapse, while prompt-induced biases remain unchanged. Through topic-level analysis (Appendix A.3), we find that question reformulation yields less improvement on economic topics compared to more politically or socially sensitive ones, yet this phenomenon is worth exploring in more depth in future work.

Our study underscores the importance of careful prompt design for LLM-based population simulations. Future work should explore hybrid approaches that combine principled prompt strategies with extensive cross-model evaluation and fine-grained question validation to better align LLM-generated responses with real-world statistics in sensitive social and political domains.

## Limitations

**Model dependence.** We observe variation in baseline silicon-sampling performance across LLMs. These differences likely reflect model-specific training data, architectures, and bias mechanisms. Although question reformulation generally reduces SDB, its effectiveness may vary in other models. Thus, our conclusions should not be assumed to generalize to all LLMs, and broader evaluation across model families is needed.

**Population coverage.** Our analysis is limited to selected demographic variables and U.S. population represented in the ANES data. Consequently, the effectiveness of mitigation strategies like question reformulation may not generalize to unexplored attributes or international contexts. Future research should validate these findings across broader demographic dimensions and global survey datasets.

**Bias vs. knowledge limitations.** Disentangling social desirability bias from insufficient population knowledge remains challenging. A model's tendency to default to a "safe" response may reflect normative pressure, lack of group-specific knowledge, or both. In cases where an LLM lacks accurate internal representations of subgroup preferences, bias mitigation alone may be ineffective or even misleading. This limits our ability to precisely quantify the magnitude of SDB and the upper bound of mitigation performance.

**Contextual independence.** Our study utilizes single-item prompting to isolate the effects of specific mitigation strategies and questions, maintaining a controlled environment. However, real-world respondents do not provide answers in isolation; they navigate a sequence of questions that are susceptible to Question Order Bias McFarland (1981). Thus, while our study establishes the efficacy of reformulation for individual items, it does not account for the cumulative contextual biases that may emerge in full-scale sequential survey simulations.

## Ethical Considerations

Silicon sampling raises ethical complications that deserve careful attention. While LLMs efficiently simulate population-level responses, their outputs are model-based, not human-based, and should not be treated as empirical facts. Researchers should clearly disclose the synthetic nature of these data and *label synthetic outputs wherever they appear.*

Bias is another concern. Even carefully framed prompts cannot eliminate the effects of societal inequalities, blind spots, or misinformation embedded in training data. This is critical when simulating perspectives from marginalized groups, where biased outputs can reinforce stigma. Transparency in prompt design, model parameters, and demographic conditioning, as well as a brief *harm review* for sensitive items, is essential. Participant privacy must also be protected. Research should rely on aggregate data, avoid reconstructing individual responses, comply with dataset licenses, and share only what is needed for replication.

Overall, this research should support human perspectives rather than replace them, with transparency, restraint, and humility as guiding principles.

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

# A    Appendix

## A.1    Implementation details

We use Llama-3.1-8B-Instruct (Llama-8B), Llama-3.1-70B-Instruct (Llama-70B) and the GPT-4.1-mini in our study. All models were configured for deterministic output to ensure consistent comparison and maximal reproducibility. For Llama-3.1-8B-Instruct, stochastic sampling was disabled (`do_sample=False`), and the maximum token limit was set to two, sufficient for the required single-word numerical answer options. For Llama-3.1-70B-Instruct, we observed frequent generation failures under standard constrained decoding. For example, the model often yielded refusals such as: "I don't have enough information to answer this question". We therefore adopt a classification-based decoding strategy: instead of generating a response, we select the answer whose corresponding token receives the highest next-token probability. This is equivalent to deterministic decoding and avoids any generation-related issues, and improves experimental run time. For GPT-4.1-mini, we followed the same generation procedure as in Llama-3.1-8B-Instruct, and the temperature was set to 0 to ensure deterministic behavior. Llama models were run on a high-performance computing cluster SURF with nodes equipped with NVIDIA H100 GPUs, and the GPT-4.1-mini model was accessed via OpenAI API.

## A.2    Additional response results for Reverse-coded, Priming and Preamble conditions

Figure 6 shows response results for different answer options across questions for the Reverse-coded condition on GPT-4.1-mini. No evidence shows that reverse-coding is an effective strategy for SDB mitigation. Figure 7 compares results on the Priming (left) and Preamble (right) conditions on Llama-70B and GPT-4.1-mini. The results overall showed decreased answer diversity and increased JS-divergence scores compared to the Replicate condition.

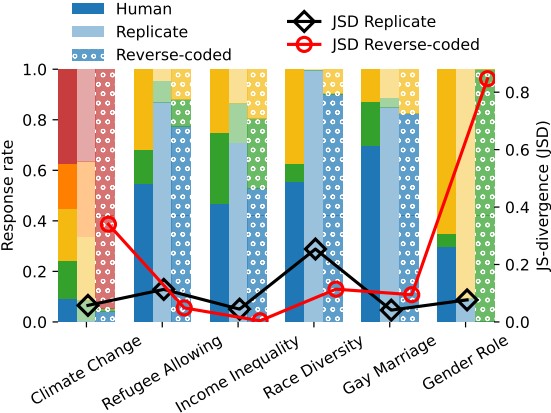

Figure 6: Human, Replicate, and Reverse-coded responses for six eligible questions on GPT-4.1-mini.

## A.3    Topic-level analysis on Reformulated condition

To understand the thematic drivers of model alignment, we categorize the survey questions into three theory-driven groups: (i) Policy and Public Safety, involving preferences for specific government actions; (ii) Identity and Social Norms, probing moral evaluations of social arrangements; and (iii) Economics, focusing on fiscal and material evaluations. This categorization is theory-driven and independent of model performance, allowing us to analyze how Social Desirability Bias (SDB) and mitigation strategies vary across topics.

Table 4 summarizes the effectiveness of question reformulation across these categories, where a question is deemed improved if at least two of three LLMs exhibit a statistically significant reduction in JS-divergence compared to the replicate condition. We exclude the other conditions, which showed no consistent improve-

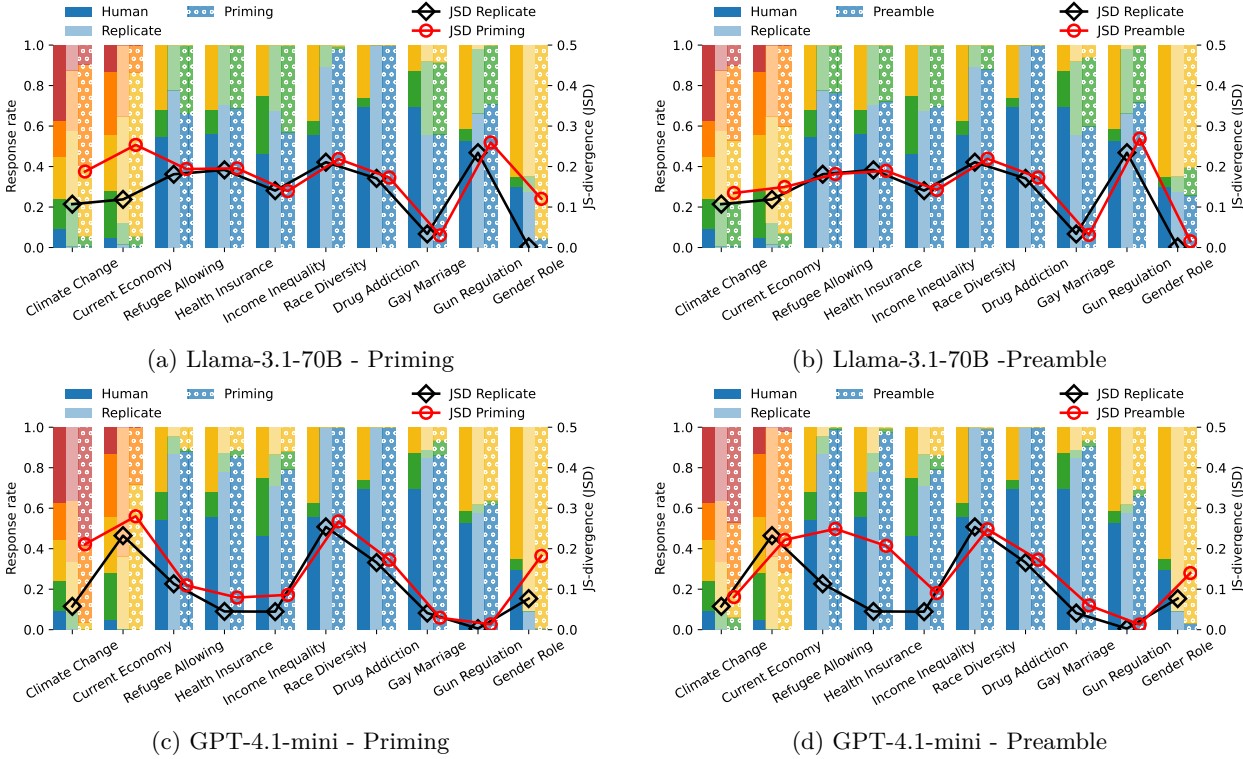

(a) Llama-3.1-70B - Priming

(b) Llama-3.1-70B - Preamble

(c) GPT-4.1-mini - Priming

(d) GPT-4.1-mini - Preamble

Figure 7: Human, Replicate, and Priming/Preamble responses for ten questions on Llama-70B and GPT-4.1-mini.

ment across models or questions. Reformulation performs most consistently for policy and public safety questions (3/4), moderately for identity and social norms (2/3), and poorly for economic topics (1/3).

Note that for *Gender Role* in the identity and social norms category, baseline (Replicate) alignment is already high for the Llama models (see Figure 2), leaving little room for further improvement and making reformulation effects harder to assess. Overall, the results suggest a tendency toward weaker gains of question reformulation on economic topics, though this pattern is not fully consistent. Future work should examine more questions across topics and finer-grained semantic properties to better understand when reformulation is most effective.

Table 4: Effectiveness of question reformulation across topic categories. A question is considered improved if at least two of three LLMs show significantly lower JS-divergence than the Replicate condition.

| Topic category | Question | # LLMs improved | Category success rate |
|---|---|---|---|
| Policy and Public Safety | Drug Addiction | 3 | 3/4 |
| | Refugee Allowing | 3 | |
| | Climate Change | 2 | |
| | Gun Regulation | 1 | |
| Identity and Social Norms | Gay Marriage | 2 | 2/3 |
| | Race Diversity | 2 | |
| | Gender Role | 1 | |
| Economic | Current Economy | 2 | 1/3 |
| | Income Inequality | 1 | |
| | Health Insurance | 1 | |

### A.4 Demographic shifts between ANES 2020 and 2024

Figure 8 presents the non-response rates across overlapping demographic axes for ANES 2020 and 2024. The meaningful demographic features are maintained, and the drop in church attendance may be due to the addition of a new option "Inapplicable" in ANES 2024.

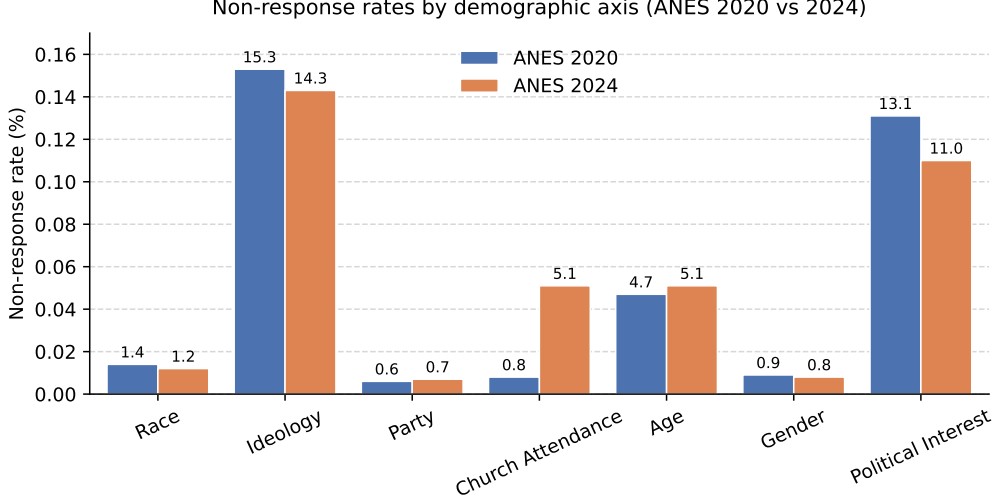

Figure 8: Comparison of non-response rates across overlapping demographic axes for ANES 2020 and 2024.

Figure 9 further illustrates detailed subgroup proportions for *Ideology*, *Party*, and *Political Interest*. We found for *Ideology*, a small portion of respondents became more conservative. For *Party*, more respondents tend to be less concentrated on the most neutral option, "Independent", and were more polarized. For *Political Interest*, relatively more respondents became less interested in 2024 compared to 2020.

### A.5 Statistical analysis and JS-divergence for ANES 2024

Table 5 provides the statistical details for the temporal analysis presented in the main paper subsection 4.4.

Table 5: Mean JS divergence and 95% bootstrap confidence intervals between ANES and silicon distributions by item and condition, using **GPT-4.1-mini** (**ANES 2024**). Results are on overlapping survey questions in ANES 2020 and 2024.

| Question | Replicate | Reformulated |
|---|---|---|
| Climate Change | 0.0866 [0.0832, 0.0904] | 0.0364 [0.0326, 0.0403] |
| Health Insurance | 0.0290 [0.0253, 0.0329] | 0.0330 [0.0293, 0.0370] |
| Income Inequality | 0.0234 [0.0203, 0.0269] | 0.1100 [0.1045, 0.1154] |
| Race Diversity | 0.2931 [0.2861, 0.3007] | 0.1827 [0.1752, 0.1902] |
| Drug Addiction | 0.1229 [0.1201, 0.1256] | 0.1176 [0.1162, 0.1189] |
| Gay Marriage | 0.0317 [0.0280, 0.0357] | 0.0056 [0.0040, 0.0073] |
| Gun Regulation | 0.0013 [0.0006, 0.0021] | 0.0172 [0.0143, 0.0201] |
| Gender Role | 0.0286 [0.0254, 0.0317] | 0.0162 [0.0135, 0.0191] |

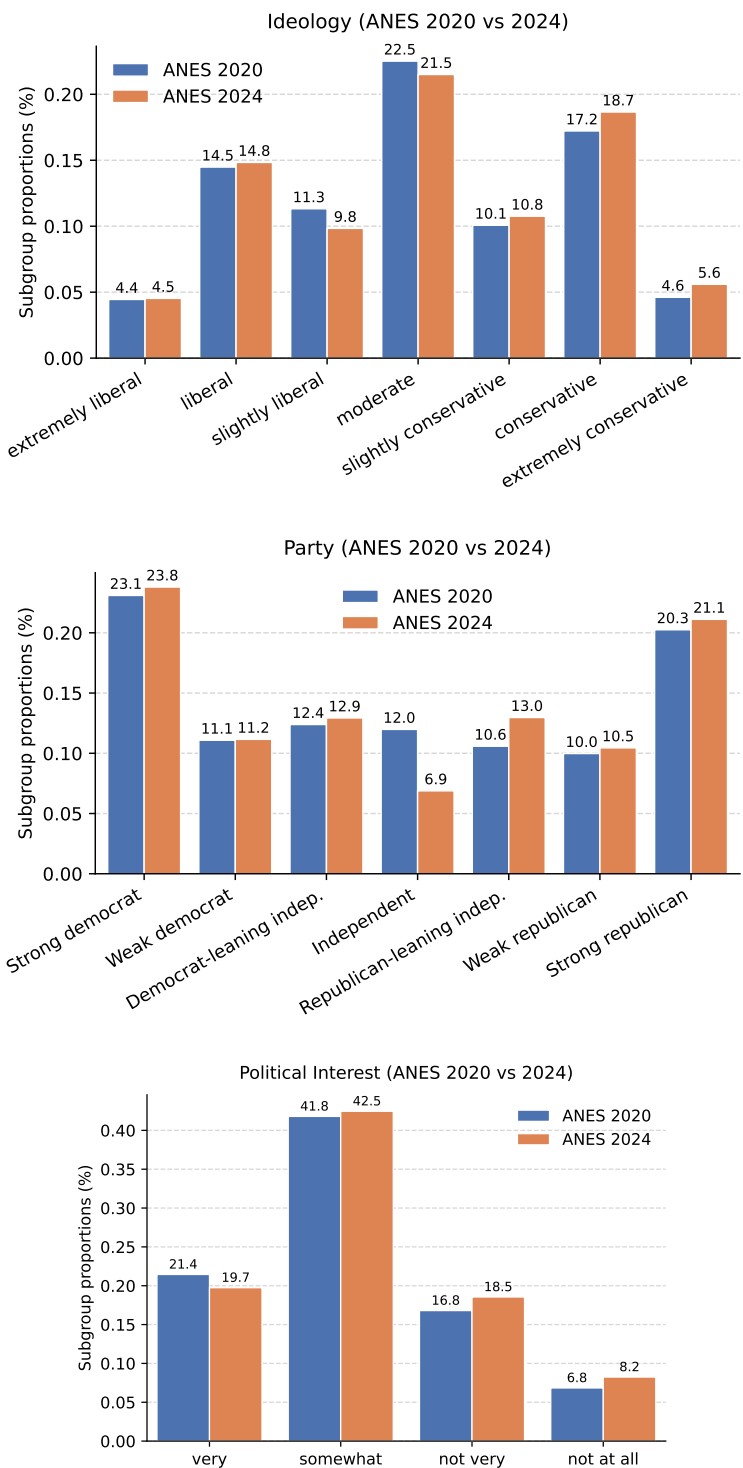

Figure 9: Subgroup proportions for Ideology, Party, and Political Interest, ANES 2020 vs. 2024.

### A.6 Full statistical analysis and JS-divergence for ANES 2020

This section presents the differences in JS-divergence scores between the Replicate condition and the four other prompting conditions, on three LLMs. Statistical significance of these differences is marked using an asterisk (∗), indicating non-overlapping bootstrap confidence intervals.

Table 6: Difference in JS-divergence between Replicate and four prompting conditions, using **Llama-8B**.

| Question | Replicate | Reformulated - Replicate | Reverse-Coded - Replicate | Priming - Replicate | Preamble - Replicate |
|---|---|---|---|---|---|
| Climate Change | 0.41 | -0.048* | -0.198* | 0.006 | 0.001 |
| Current Economy | 0.309 | -0.189* | N/A | 0.057* | -0.016* |
| Refugee Allowing | 0.275 | -0.065* | 0.032* | 0.0 | -0.026* |
| Health Insurance | 0.263 | -0.049* | N/A | 0.002 | 0.002 |
| Income Inequality | 0.197 | -0.049* | -0.134* | 0.074* | 0.017* |
| Race Diversity | 0.187 | -0.047* | 0.068* | -0.154* | -0.177* |
| Drug Addiction | 0.173 | -0.013* | N/A | 0.0 | -0.005* |
| Gay Marriage | 0.143 | -0.09* | 0.029* | 0.029* | 0.029* |
| Gun Regulation | 0.119 | -0.055* | N/A | 0.096* | 0.085* |
| Gender Role | 0.016 | 0.142* | 0.793* | 0.172* | 0.139* |

Table 7: Mean JS divergence and 95% bootstrap confidence intervals between ANES and silicon distributions by item and condition, using **Llama-8B**.

| Question | Replicate | Reformulated | Reverse-Coded | Priming | Preamble |
|---|---|---|---|---|---|
| Climate Change | 0.4103 [0.4051, 0.4155] | 0.3621 [0.3568, 0.3674] | 0.2129 [0.2072, 0.2187] | 0.4164 [0.4151, 0.4177] | 0.4117 [0.4077, 0.4156] |
| Current Economy | 0.3089 [0.3044, 0.3135] | 0.1201 [0.1171, 0.1234] | N/A | 0.3664 [0.3600, 0.3730] | 0.2930 [0.2874, 0.2983] |
| Refugee Allowing | 0.2755 [0.2735, 0.2772] | 0.2103 [0.2070, 0.2139] | 0.3077 [0.3008, 0.3149] | 0.2755 [0.2735, 0.2772] | 0.2499 [0.2450, 0.2548] |
| Health Insurance | 0.2627 [0.2597, 0.2651] | 0.2137 [0.2102, 0.2172] | N/A | 0.2652 [0.2637, 0.2660] | 0.2650 [0.2633, 0.2660] |
| Income Inequality | 0.1976 [0.1890, 0.2063] | 0.1489 [0.1446, 0.1531] | 0.0635 [0.0581, 0.0688] | 0.2718 [0.2644, 0.2799] | 0.2142 [0.2054, 0.2227] |
| Race Diversity | 0.1867 [0.1793, 0.1942] | 0.1400 [0.1331, 0.1469] | 0.2542 [0.2494, 0.2589] | 0.0326 [0.0306, 0.0346] | 0.0101 [0.0080, 0.0124] |
| Drug Addiction | 0.1727 [0.1727, 0.1727] | 0.1603 [0.1559, 0.1644] | N/A | 0.1727 [0.1727, 0.1727] | 0.1681 [0.1650, 0.1709] |
| Gay Marriage | 0.1432 [0.1391, 0.1476] | 0.0527 [0.0483, 0.0572] | 0.1721 [0.1721, 0.1721] | 0.1721 [0.1721, 0.1721] | 0.1721 [0.1721, 0.1721] |
| Gun Regulation | 0.1196 [0.1125, 0.1267] | 0.0643 [0.0588, 0.0699] | N/A | 0.2156 [0.2071, 0.2243] | 0.2044 [0.1952, 0.2130] |
| Gender Role | 0.0162 [0.0136, 0.0191] | 0.1586 [0.1503, 0.1669] | 0.8097 [0.8016, 0.8178] | 0.1887 [0.1840, 0.1935] | 0.1556 [0.1491, 0.1625] |

Table 8: Difference in JS-divergence between Replicate and four prompting conditions, using **Llama-70B**.

| Question | Replicate | Reformulated - Replicate | Reverse-Coded - Replicate | Priming - Replicate | Preamble - Replicate |
|---|---|---|---|---|---|
| Climate Change | 0.107 | 0.049* | -0.066* | 0.081* | 0.028* |
| Current Economy | 0.119 | 0.126* | N/A | 0.135* | 0.031* |
| Refugee Allowing | 0.181 | -0.036* | 0.008* | 0.013* | 0.0 |
| Health Insurance | 0.192 | 0.001 | N/A | 0.004 | -0.002 |
| Income Inequality | 0.141 | 0.0 | -0.004 | -0.001 | 0.002 |
| Race Diversity | 0.211 | 0.042* | 0.037* | 0.008 | 0.009* |
| Drug Addiction | 0.171 | -0.02* | N/A | 0.002 | 0.002 |
| Gay Marriage | 0.034 | 0.051* | 0.072* | -0.003 | -0.002 |
| Gun Regulation | 0.234 | 0.068* | N/A | 0.025* | 0.035* |
| Gender Role | 0.002 | 0.188* | 0.756* | 0.119* | 0.014* |

Table 9: Mean JS divergence and 95% bootstrap confidence intervals between ANES and silicon distributions by item and condition, using **Llama-70B**.

| Question | Replicate | Reformulated | Reverse-Coded | Priming | Preamble |
|---|---|---|---|---|---|
| Climate Change | 0.1073 [0.1009, 0.1141] | 0.1566 [0.1491, 0.1640] | 0.0413 [0.0370, 0.0456] | 0.1879 [0.1792, 0.1966] | 0.1350 [0.1283, 0.1418] |
| Current Economy | 0.1191 [0.1145, 0.1242] | 0.2452 [0.2370, 0.2534] | N/A | 0.2537 [0.2455, 0.2622] | 0.1497 [0.1443, 0.1554] |
| Refugee Allowing | 0.1810 [0.1807, 0.1815] | 0.1449 [0.1389, 0.1511] | 0.1892 [0.1875, 0.1912] | 0.1939 [0.1918, 0.1963] | 0.1815 [0.1810, 0.1822] |
| Health Insurance | 0.1918 [0.1897, 0.1937] | 0.1930 [0.1909, 0.1953] | N/A | 0.1955 [0.1933, 0.1979] | 0.1894 [0.1877, 0.1913] |
| Income Inequality | 0.1405 [0.1397, 0.1416] | 0.1406 [0.1397, 0.1417] | 0.1370 [0.1340, 0.1397] | 0.1394 [0.1373, 0.1412] | 0.1424 [0.1412, 0.1438] |
| Race Diversity | 0.2109 [0.2066, 0.2148] | 0.2533 [0.2489, 0.2577] | 0.2476 [0.2446, 0.2505] | 0.2187 [0.2121, 0.2252] | 0.2193 [0.2192, 0.2197] |
| Drug Addiction | 0.1708 [0.1690, 0.1727] | 0.1510 [0.1494, 0.1528] | N/A | 0.1727 [0.1727, 0.1727] | 0.1727 [0.1727, 0.1727] |
| Gay Marriage | 0.0337 [0.0298, 0.0378] | 0.0846 [0.0782, 0.0909] | 0.1060 [0.1038, 0.1084] | 0.0302 [0.0265, 0.0341] | 0.0314 [0.0276, 0.0353] |
| Gun Regulation | 0.2342 [0.2263, 0.2423] | 0.3023 [0.2980, 0.3066] | N/A | 0.2595 [0.2539, 0.2653] | 0.2689 [0.2649, 0.2727] |
| Gender Role | 0.0017 [0.0009, 0.0028] | 0.1894 [0.1803, 0.1987] | 0.7577 [0.7499, 0.7656] | 0.1202 [0.1141, 0.1266] | 0.0154 [0.0128, 0.0182] |

Table 10: Difference in JS-divergence between Replicate and four prompting conditions, using **GPT-4.1-mini**.

| Question | Replicate | Reformulated - Replicate | Reverse-Coded - Replicate | Priming - Replicate | Preamble - Replicate |
|---|---|---|---|---|---|
| Climate Change | 0.058 | -0.032* | 0.283* | 0.153* | 0.023* |
| Current Economy | 0.232 | -0.06* | N/A | 0.048* | -0.011 |
| Refugee Allowing | 0.113 | -0.086* | -0.064* | -0.004 | 0.136* |
| Health Insurance | 0.045 | 0.026* | N/A | 0.034* | 0.162* |
| Income Inequality | 0.044 | 0.06* | -0.04* | 0.042* | 0.046* |
| Race Diversity | 0.254 | -0.068* | -0.14* | 0.013* | -0.006 |
| Drug Addiction | 0.166 | -0.022* | N/A | 0.007* | 0.007* |
| Gay Marriage | 0.041 | -0.009* | 0.054* | -0.012* | 0.019* |
| Gun Regulation | 0.003 | 0.012* | N/A | 0.011* | 0.01* |
| Gender Role | 0.077 | -0.067* | 0.77* | 0.106* | 0.063* |

Table 11: Mean JS divergence and 95% bootstrap confidence intervals between ANES and silicon distributions by item and condition, using **GPT-4.1-mini**.

| Question | Replicate | Reformulated | Reverse-Coded | Priming | Preamble |
|---|---|---|---|---|---|
| Climate Change | 0.0578 [0.0537, 0.0622] | 0.0261 [0.0228, 0.0296] | 0.3404 [0.3343, 0.3467] | 0.2109 [0.2053, 0.2166] | 0.0807 [0.0761, 0.0857] |
| Current Economy | 0.2324 [0.2268, 0.2378] | 0.1725 [0.1642, 0.1808] | N/A | 0.2801 [0.2767, 0.2837] | 0.2219 [0.2161, 0.2277] |
| Refugee Allowing | 0.1134 [0.1069, 0.1202] | 0.0269 [0.0233, 0.0304] | 0.0489 [0.0441, 0.0537] | 0.1095 [0.1027, 0.1161] | 0.2490 [0.2429, 0.2546] |
| Health Insurance | 0.0452 [0.0407, 0.0498] | 0.0708 [0.0653, 0.0764] | N/A | 0.0794 [0.0735, 0.0853] | 0.2073 [0.2005, 0.2143] |
| Income Inequality | 0.0445 [0.0399, 0.0492] | 0.1045 [0.0990, 0.1097] | 0.0041 [0.0028, 0.0056] | 0.0861 [0.0801, 0.0921] | 0.0905 [0.0844, 0.0971] |
| Race Diversity | 0.2540 [0.2490, 0.2589] | 0.1863 [0.1797, 0.1928] | 0.1141 [0.1074, 0.1213] | 0.2674 [0.2674, 0.2674] | 0.2479 [0.2421, 0.2531] |
| Drug Addiction | 0.1660 [0.1636, 0.1682] | 0.1441 [0.1439, 0.1444] | N/A | 0.1727 [0.1727, 0.1727] | 0.1727 [0.1727, 0.1727] |
| Gay Marriage | 0.0412 [0.0370, 0.0456] | 0.0320 [0.0283, 0.0360] | 0.0949 [0.0945, 0.0954] | 0.0296 [0.0259, 0.0334] | 0.0600 [0.0548, 0.0650] |
| Gun Regulation | 0.0028 [0.0017, 0.0040] | 0.0150 [0.0124, 0.0179] | N/A | 0.0134 [0.0110, 0.0159] | 0.0129 [0.0105, 0.0156] |
| Gender Role | 0.0770 [0.0712, 0.0830] | 0.0103 [0.0081, 0.0127] | 0.8471 [0.8471, 0.8471] | 0.1823 [0.1772, 0.1875] | 0.1400 [0.1337, 0.1463] |

### A.7 Full stratified JS-divergence analysis by demographic subgroup

We present the full demographic subgroup analysis of the change (Δ) in JS-divergence scores. The results compare each mitigation prompting condition against the Replicate condition for all three LLMs. Note for each tables, questions are sorted horizontally (left to right) based on the overall average difference in JS-divergence, from low to high.

Table 12: Difference in JS-divergence (Δ) between **Reformulated** and Replicate conditions on stratified subgroups using **Llama-8B**. The color gradient indicates the magnitude of the change: Darker green signifies greater improvement in alignment (Δ < 0), while darker red signifies greater worsening of alignment (Δ > 0).

| | | Current Economy | Gay Marriage | Refugee Allowing | Gun Regulation | Income Inequality | Health Insurance | Climate Change | Race Diversity | Drug Addiction | Gender Role | Average |
|---|---|---|---|---|---|---|---|---|---|---|---|---|
| Race | White | -0.204 | -0.086 | -0.071 | -0.055 | -0.066 | -0.053 | -0.049 | -0.055 | -0.009 | 0.116 | -0.053 |
| | Black | -0.124 | -0.125 | -0.046 | -0.090 | 0.015 | -0.039 | -0.037 | -0.023 | -0.031 | 0.288 | -0.021 |
| | Asian | -0.182 | -0.083 | -0.042 | -0.022 | -0.005 | -0.066 | -0.052 | 0.012 | -0.018 | 0.221 | -0.024 |
| | Native American | -0.134 | -0.050 | -0.043 | -0.040 | 0.031 | -0.036 | -0.014 | 0.005 | -0.018 | 0.201 | -0.010 |
| | Hispanic | -0.183 | -0.117 | -0.053 | -0.074 | -0.024 | 0.015 | -0.044 | -0.016 | 0.000 | 0.260 | -0.024 |
| Discuss Politics | Like to discuss | -0.181 | -0.072 | -0.064 | -0.055 | -0.053 | -0.048 | -0.050 | -0.086 | -0.009 | 0.159 | -0.046 |
| | Never discuss | -0.201 | -0.147 | -0.097 | -0.060 | -0.035 | -0.038 | -0.052 | -0.068 | -0.053 | 0.085 | -0.067 |
| Ideology | Extremely liberal | 0.000 | 0.000 | 0.000 | 0.064 | 0.000 | 0.000 | 0.043 | 0.000 | 0.000 | 0.274 | 0.038 |
| | Liberal | -0.121 | 0.000 | 0.000 | -0.041 | -0.004 | 0.000 | 0.011 | -0.008 | 0.000 | 0.074 | -0.009 |
| | Slightly liberal | -0.116 | -0.020 | 0.000 | -0.073 | -0.035 | 0.000 | 0.039 | 0.012 | -0.006 | 0.067 | -0.013 |
| | Moderate | -0.092 | -0.028 | 0.000 | -0.024 | -0.066 | 0.000 | 0.037 | -0.018 | -0.018 | 0.095 | -0.011 |
| | Slightly cons. | -0.281 | -0.168 | -0.007 | -0.110 | -0.184 | -0.007 | -0.040 | -0.013 | -0.018 | 0.274 | -0.056 |
| | Conservative | -0.328 | -0.243 | -0.128 | -0.057 | -0.156 | -0.117 | -0.161 | -0.077 | -0.012 | 0.228 | -0.105 |
| | Extremely cons. | -0.071 | -0.073 | -0.256 | -0.060 | 0.008 | -0.286 | 0.005 | -0.172 | -0.024 | 0.205 | -0.072 |
| Party | Strong democrat | -0.049 | -0.003 | 0.000 | 0.032 | -0.025 | 0.000 | -0.003 | -0.042 | 0.000 | 0.189 | 0.010 |
| | Weak democrat | -0.077 | -0.083 | -0.036 | -0.035 | -0.028 | -0.042 | -0.004 | -0.036 | 0.000 | 0.063 | -0.028 |
| | Democrat-leaning indep. | -0.063 | -0.035 | 0.000 | 0.010 | -0.023 | 0.000 | 0.027 | -0.039 | 0.000 | 0.000 | -0.012 |
| | Independent | -0.038 | -0.086 | -0.059 | -0.028 | -0.001 | -0.047 | -0.048 | -0.079 | -0.047 | 0.002 | -0.043 |
| | Republican-leaning indep. | -0.169 | -0.151 | -0.084 | -0.083 | -0.180 | -0.072 | -0.091 | -0.073 | -0.022 | 0.222 | -0.070 |
| | Weak republican | -0.052 | -0.104 | -0.074 | -0.105 | -0.119 | -0.068 | -0.102 | -0.061 | -0.007 | 0.193 | -0.050 |
| | Strong republican | -0.403 | -0.155 | -0.181 | -0.048 | -0.107 | -0.119 | -0.109 | -0.021 | -0.007 | 0.194 | -0.096 |
| Church | Attend church | -0.206 | -0.150 | -0.065 | -0.072 | -0.072 | -0.048 | -0.052 | -0.050 | -0.015 | 0.129 | -0.060 |
| | Does not attend | -0.172 | -0.032 | -0.063 | -0.044 | -0.023 | -0.045 | -0.043 | -0.040 | -0.011 | 0.152 | -0.032 |
| Gender | Man | -0.206 | -0.102 | -0.078 | -0.061 | -0.059 | -0.056 | -0.056 | -0.038 | -0.007 | 0.153 | -0.051 |
| | Woman | -0.173 | -0.080 | -0.054 | -0.049 | -0.041 | -0.044 | -0.043 | -0.055 | -0.017 | 0.135 | -0.042 |
| Political Interest | Very | -0.211 | -0.061 | -0.075 | -0.054 | -0.082 | -0.040 | -0.043 | -0.043 | 0.000 | 0.309 | -0.027 |
| | Somewhat | -0.198 | -0.093 | -0.057 | -0.039 | -0.072 | -0.038 | -0.027 | 0.011 | 0.000 | 0.208 | -0.031 |
| | Not very | -0.140 | -0.108 | -0.054 | -0.106 | -0.033 | -0.068 | -0.029 | -0.061 | -0.030 | -0.013 | -0.064 |
| | Not at all | -0.170 | -0.093 | -0.071 | -0.109 | -0.015 | -0.080 | -0.049 | -0.199 | -0.057 | -0.043 | -0.088 |

Table 13: Difference in JS-divergence (Δ) between **Reverse-coded** and Replicate conditions on stratified subgroups using **Llama-8B**. The color gradient indicates the magnitude of the change: Darker green signifies greater improvement in alignment (Δ < 0), while darker red signifies greater worsening of alignment (Δ > 0).

| | | Climate Change | Income Inequality | Gay Marriage | Race Diversity | Refugee Allowing | Gender Role | Average |
|---|---|---|---|---|---|---|---|---|
| Race | White | -0.195 | -0.161 | 0.026 | 0.084 | 0.042 | 0.822 | 0.096 |
| | Black | -0.18 | -0.036 | 0.059 | 0.03 | -0.027 | 0.733 | 0.098 |
| | Asian | -0.231 | -0.071 | 0.021 | 0.002 | 0.052 | 0.717 | 0.08 |
| | Native American | -0.184 | 0.03 | 0.034 | 0.021 | 0.074 | 0.701 | 0.098 |
| | Hispanic | -0.138 | -0.121 | 0.047 | -0.031 | -0.073 | 0.732 | 0.074 |
| Discuss Politics | Like to discuss politics | -0.22 | -0.13 | 0.023 | 0.019 | 0.039 | 0.797 | 0.089 |
| | Never discuss politics | 0.03 | -0.195 | 0.036 | 0.165 | -0.078 | 0.785 | 0.126 |
| Ideology | Extremely liberal | -0.623 | 0.754 | 0.0 | 0.0 | 0.783 | 0.715 | 0.285 |
| | Liberal | -0.386 | 0.649 | 0.0 | 0.0 | 0.747 | 0.718 | 0.29 |
| | Slightly liberal | -0.414 | 0.125 | 0.0 | 0.025 | 0.393 | 0.724 | 0.135 |
| | Moderate | -0.14 | -0.209 | 0.0 | 0.028 | 0.108 | 0.799 | 0.116 |
| | Slightly conservative | 0.07 | -0.273 | 0.0 | 0.127 | -0.098 | 0.849 | 0.088 |
| | Conservative | -0.043 | 0.118 | 0.017 | 0.132 | -0.202 | 0.882 | 0.121 |
| | Extremely conservative | -0.18 | 0.53 | 0.312 | 0.243 | -0.063 | 0.563 | 0.175 |
| Party | Strong democrat | -0.453 | 0.299 | 0.0 | 0.006 | 0.578 | 0.656 | 0.196 |
| | Weak democrat | -0.306 | 0.065 | 0.0 | 0.072 | 0.249 | 0.584 | 0.106 |
| | Democrat-leaning indep. | -0.361 | 0.187 | 0.0 | 0.029 | 0.459 | 0.659 | 0.158 |
| | Independent | -0.024 | -0.111 | 0.023 | 0.053 | 0.097 | 0.776 | 0.138 |
| | Republican-leaning indep. | 0.054 | -0.295 | 0.019 | 0.091 | -0.15 | 0.855 | 0.087 |
| | Weak republican | 0.054 | -0.273 | 0.062 | 0.1 | -0.079 | 0.854 | 0.109 |
| | Strong republican | -0.037 | 0.069 | 0.076 | 0.137 | -0.186 | 0.827 | 0.129 |
| Church | Attend church | -0.143 | -0.198 | 0.039 | 0.053 | -0.002 | 0.802 | 0.091 |
| | Does not attend church | -0.242 | -0.067 | 0.019 | 0.081 | 0.063 | 0.787 | 0.099 |
| Gender | Man | -0.214 | -0.146 | 0.029 | 0.074 | 0.005 | 0.773 | 0.084 |
| | Woman | -0.183 | -0.123 | 0.029 | 0.06 | 0.055 | 0.813 | 0.103 |
| Political Interest | Very | -0.367 | -0.068 | 0.024 | 0.041 | 0.077 | 0.784 | 0.078 |
| | Somewhat | -0.217 | -0.108 | 0.025 | 0.071 | 0.01 | 0.81 | 0.101 |
| | Not very | -0.054 | -0.217 | 0.027 | 0.079 | 0.047 | 0.782 | 0.103 |
| | Not at all | 0.007 | -0.154 | 0.044 | 0.125 | -0.016 | 0.698 | 0.103 |

Table 14: Difference in JS-divergence (Δ) between **Priming** and Replicate conditions on stratified subgroups using **Llama-8B**. The color gradient indicates the magnitude of the change: Darker green signifies greater improvement in alignment (Δ < 0), while darker red signifies greater worsening of alignment (Δ > 0).

| | | Race Diversity | Refugee Allowing | Drug Addiction | Climate Change | Health Insurance | Gay Marriage | Current Economy | Income Inequality | Gun Regulation | Gender Role | Average |
|---|---|---|---|---|---|---|---|---|---|---|---|---|
| Race | White | -0.141 | 0.002 | 0.0 | 0.01 | 0.0 | 0.026 | 0.057 | 0.074 | 0.069 | 0.174 | 0.027 |
| | Black | -0.205 | -0.014 | 0.0 | -0.0 | 0.005 | 0.059 | 0.034 | 0.031 | 0.253 | 0.175 | 0.034 |
| | Asian | -0.222 | 0.0 | 0.0 | -0.011 | 0.0 | 0.021 | 0.068 | 0.111 | 0.14 | 0.169 | 0.028 |
| | Native American | -0.136 | 0.0 | 0.0 | 0.003 | 0.0 | 0.034 | 0.031 | 0.105 | 0.212 | 0.168 | 0.042 |
| | Hispanic | -0.218 | 0.0 | 0.0 | -0.003 | 0.052 | 0.047 | 0.014 | 0.155 | 0.118 | 0.193 | 0.036 |
| Discuss Politics | Like to discuss politics | -0.203 | -0.002 | 0.0 | 0.001 | 0.003 | 0.023 | 0.054 | 0.064 | 0.081 | 0.173 | 0.019 |
| | Never discuss politics | -0.149 | 0.0 | 0.0 | 0.003 | 0.0 | 0.036 | 0.098 | 0.085 | 0.125 | 0.179 | 0.038 |
| Ideology | Extremely liberal | -0.011 | 0.0 | 0.0 | -0.01 | 0.0 | 0.0 | -0.014 | 0.0 | 0.102 | 0.051 | 0.012 |
| | Liberal | -0.005 | 0.0 | 0.0 | -0.009 | 0.0 | 0.0 | 0.034 | 0.0 | 0.206 | 0.044 | 0.027 |
| | Slightly liberal | -0.074 | 0.0 | 0.0 | -0.026 | 0.0 | 0.0 | 0.078 | 0.0 | 0.271 | 0.041 | 0.029 |
| | Moderate | -0.194 | 0.0 | 0.0 | -0.033 | 0.0 | 0.0 | 0.054 | 0.0 | 0.083 | 0.12 | 0.003 |
| | Slightly conservative | -0.142 | 0.0 | 0.0 | -0.03 | 0.0 | 0.0 | 0.06 | 0.134 | -0.028 | 0.203 | 0.02 |
| | Conservative | -0.21 | 0.0 | 0.0 | 0.038 | -0.0 | 0.017 | 0.018 | 0.188 | 0.007 | 0.293 | 0.035 |
| | Extremely conservative | -0.127 | -0.0 | 0.0 | 0.169 | 0.043 | 0.312 | 0.042 | 0.209 | 0.067 | 0.253 | 0.097 |
| Party | Strong democrat | -0.068 | 0.0 | 0.0 | -0.0 | 0.0 | 0.0 | 0.011 | 0.0 | 0.205 | 0.043 | 0.019 |
| | Weak democrat | -0.091 | 0.0 | 0.0 | -0.031 | 0.0 | 0.0 | 0.074 | 0.04 | 0.104 | 0.048 | 0.014 |
| | Democrat-leaning indep. | -0.094 | 0.0 | 0.0 | -0.033 | 0.0 | 0.0 | 0.057 | 0.016 | 0.14 | 0.01 | 0.01 |
| | Independent | -0.212 | 0.0 | 0.0 | -0.009 | 0.007 | 0.023 | 0.049 | 0.078 | 0.101 | 0.142 | 0.018 |
| | Republican-leaning indep. | -0.182 | 0.0 | 0.0 | 0.005 | 0.0 | 0.019 | 0.041 | 0.087 | 0.013 | 0.216 | 0.02 |
| | Weak republican | -0.211 | 0.0 | 0.0 | -0.018 | 0.0 | 0.062 | 0.111 | 0.102 | 0.039 | 0.172 | 0.026 |
| | Strong republican | -0.264 | -0.0 | 0.0 | 0.053 | 0.004 | 0.076 | 0.094 | 0.177 | 0.028 | 0.218 | 0.039 |
| Church | Attend church | -0.187 | 0.0 | 0.0 | 0.009 | 0.002 | 0.039 | 0.061 | 0.065 | 0.117 | 0.203 | 0.031 |
| | Does not attend church | -0.125 | 0.0 | 0.0 | 0.004 | 0.002 | 0.019 | 0.052 | 0.083 | 0.08 | 0.145 | 0.026 |
| Gender | Man | -0.147 | 0.0 | 0.0 | 0.009 | 0.002 | 0.029 | 0.066 | 0.057 | 0.09 | 0.181 | 0.029 |
| | Woman | -0.161 | -0.0 | 0.0 | 0.003 | 0.002 | 0.029 | 0.051 | 0.09 | 0.1 | 0.165 | 0.028 |
| Political Interest | Very | -0.115 | -0.004 | 0.0 | 0.012 | 0.003 | 0.024 | 0.069 | 0.047 | 0.067 | 0.159 | 0.026 |
| | Somewhat | -0.161 | 0.002 | 0.0 | -0.011 | 0.002 | 0.025 | 0.088 | 0.053 | 0.116 | 0.166 | 0.028 |
| | Not very | -0.165 | 0.0 | 0.0 | 0.006 | 0.0 | 0.027 | 0.033 | 0.087 | 0.102 | 0.171 | 0.026 |
| | Not at all | -0.211 | 0.0 | 0.0 | 0.029 | 0.023 | 0.044 | 0.012 | 0.125 | 0.13 | 0.171 | 0.032 |

Table 15: Difference in JS-divergence (Δ) between **Preamble** and Replicate conditions on stratified subgroups using **Llama-8B**. The color gradient indicates the magnitude of the change: Darker green signifies greater improvement in alignment (Δ < 0), while darker red signifies greater worsening of alignment (Δ > 0).

| | | Race Diversity | Refugee Allowing | Current Economy | Drug Addiction | Climate Change | Health Insurance | Income Inequality | Gay Marriage | Gun Regulation | Gender Role | Average |
|---|---|---|---|---|---|---|---|---|---|---|---|---|
| Race | White | -0.166 | -0.016 | -0.014 | 0.0 | 0.007 | 0.0 | 0.023 | 0.026 | 0.081 | 0.142 | 0.008 |
| | Black | -0.244 | -0.063 | -0.053 | -0.018 | -0.013 | 0.004 | 0.017 | 0.059 | 0.112 | 0.136 | -0.006 |
| | Asian | -0.189 | -0.034 | -0.01 | -0.008 | -0.019 | 0.0 | -0.009 | 0.021 | 0.108 | 0.162 | 0.002 |
| | Native American | -0.136 | -0.034 | -0.045 | -0.019 | -0.015 | 0.0 | 0.008 | 0.034 | 0.102 | 0.088 | -0.002 |
| | Hispanic | -0.191 | -0.059 | -0.017 | -0.024 | -0.006 | 0.052 | 0.019 | 0.047 | 0.05 | 0.206 | 0.008 |
| Discuss Politics | Like to discuss politics | -0.227 | -0.014 | -0.017 | -0.002 | -0.002 | 0.003 | 0.008 | 0.023 | 0.081 | 0.134 | -0.001 |
| | Never discuss politics | -0.183 | -0.115 | 0.017 | -0.028 | -0.009 | 0.0 | -0.016 | 0.036 | 0.08 | 0.143 | -0.008 |
| Ideology | Extremely liberal | -0.033 | 0.0 | -0.013 | 0.0 | 0.023 | 0.0 | 0.0 | 0.0 | 0.076 | 0.001 | 0.005 |
| | Liberal | -0.057 | 0.0 | -0.018 | 0.0 | -0.009 | 0.0 | 0.0 | 0.0 | 0.151 | 0.005 | 0.007 |
| | Slightly liberal | -0.085 | -0.014 | 0.004 | 0.0 | -0.041 | 0.0 | 0.0 | 0.0 | 0.136 | 0.01 | 0.001 |
| | Moderate | -0.133 | -0.008 | -0.001 | 0.0 | -0.019 | 0.0 | -0.007 | 0.0 | 0.148 | 0.105 | 0.009 |
| | Slightly conservative | -0.103 | -0.017 | -0.041 | 0.0 | -0.021 | 0.0 | 0.047 | 0.0 | -0.012 | 0.299 | 0.015 |
| | Conservative | -0.287 | -0.048 | -0.052 | 0.0 | 0.017 | -0.001 | 0.076 | 0.017 | -0.005 | 0.301 | 0.002 |
| | Extremely conservative | -0.106 | -0.097 | -0.007 | -0.013 | 0.095 | 0.043 | 0.035 | 0.312 | 0.011 | 0.107 | 0.038 |
| Party | Strong democrat | -0.094 | 0.0 | -0.023 | 0.0 | -0.001 | 0.0 | 0.0 | 0.0 | 0.213 | 0.0 | 0.009 |
| | Weak democrat | -0.09 | -0.03 | -0.017 | 0.0 | -0.035 | 0.0 | -0.024 | 0.0 | 0.07 | 0.01 | -0.012 |
| | Democrat-leaning indep. | -0.042 | 0.0 | -0.014 | 0.0 | -0.021 | 0.0 | -0.01 | 0.0 | 0.1 | -0.023 | -0.001 |
| | Independent | -0.179 | -0.006 | 0.001 | -0.011 | -0.012 | 0.007 | 0.053 | 0.023 | 0.087 | 0.1 | 0.006 |
| | Republican-leaning indep. | -0.166 | -0.012 | -0.013 | -0.007 | -0.002 | 0.0 | 0.034 | 0.019 | 0.007 | 0.252 | 0.011 |
| | Weak republican | -0.214 | -0.007 | -0.008 | 0.0 | -0.016 | 0.0 | 0.075 | 0.062 | 0.044 | 0.222 | 0.016 |
| | Strong republican | -0.319 | -0.078 | -0.041 | -0.007 | 0.027 | 0.008 | 0.045 | 0.076 | 0.021 | 0.244 | -0.003 |
| Church | Attend church | -0.212 | -0.024 | -0.016 | -0.008 | 0.007 | 0.002 | 0.007 | 0.039 | 0.118 | 0.162 | 0.008 |
| | Does not attend church | -0.145 | -0.026 | -0.016 | -0.002 | -0.004 | 0.002 | 0.027 | 0.019 | 0.061 | 0.12 | 0.004 |
| Gender | Man | -0.17 | -0.033 | -0.011 | -0.002 | -0.001 | 0.004 | -0.001 | 0.029 | 0.078 | 0.141 | 0.003 |
| | Woman | -0.182 | -0.018 | -0.02 | -0.007 | 0.003 | -0.0 | 0.034 | 0.029 | 0.09 | 0.139 | 0.007 |
| Political Interest | Very | -0.141 | -0.008 | -0.008 | 0.0 | 0.003 | 0.003 | 0.024 | 0.024 | 0.047 | 0.159 | 0.01 |
| | Somewhat | -0.163 | -0.03 | -0.002 | 0.0 | -0.01 | 0.002 | 0.015 | 0.025 | 0.102 | 0.151 | 0.009 |
| | Not very | -0.195 | -0.015 | -0.019 | -0.015 | 0.018 | 0.0 | 0.014 | 0.027 | 0.14 | 0.123 | 0.008 |
| | Not at all | -0.212 | -0.039 | -0.082 | -0.017 | 0.018 | 0.012 | 0.004 | 0.044 | 0.141 | 0.161 | 0.003 |

Table 16: Difference in JS-divergence ($\Delta$) between **Reformulated** and Replicate conditions on stratified subgroups using **Llama-70B**. The color gradient indicates the magnitude of the change: Darker green signifies greater improvement in alignment ($\Delta < 0$), while darker red signifies greater worsening of alignment ($\Delta > 0$).

| | | Refugee Allowing | Drug Addiction | Income Inequality | Health Insurance | Race Diversity | Climate Change | Gun Regulation | Gay Marriage | Current Economy | Gender Role | Average |
|---|---|---|---|---|---|---|---|---|---|---|---|---|
| Race | White | -0.029 | -0.019 | -0.001 | 0.005 | 0.06 | 0.052 | 0.074 | 0.053 | 0.127 | 0.203 | 0.052 |
| | Black | -0.048 | -0.023 | 0.002 | -0.012 | -0.026 | 0.068 | 0.005 | 0.005 | 0.074 | 0.139 | 0.018 |
| | Asian | -0.048 | -0.026 | 0.0 | -0.002 | 0.008 | 0.038 | 0.035 | 0.069 | 0.128 | 0.168 | 0.037 |
| | Native American | -0.044 | -0.004 | -0.013 | -0.003 | 0.038 | 0.038 | 0.075 | 0.069 | 0.202 | 0.109 | 0.047 |
| | Hispanic | -0.032 | -0.017 | 0.014 | -0.003 | 0.004 | 0.048 | 0.019 | -0.028 | 0.069 | 0.223 | 0.03 |
| Discuss Politics | Like to discuss politics | -0.032 | -0.02 | -0.001 | 0.004 | 0.048 | 0.049 | 0.068 | 0.056 | 0.138 | 0.19 | 0.05 |
| | Never discuss politics | -0.11 | -0.023 | 0.002 | -0.001 | -0.031 | 0.029 | 0.111 | 0.031 | 0.14 | 0.123 | 0.027 |
| Ideology | Extremely liberal | 0.0 | 0.0 | 0.0 | 0.0 | 0.0 | 0.024 | 0.007 | 0.004 | 0.008 | 0.423 | 0.047 |
| | Liberal | 0.0 | 0.0 | 0.0 | 0.011 | 0.0 | 0.113 | -0.008 | 0.022 | 0.158 | 0.612 | 0.091 |
| | Slightly liberal | -0.009 | 0.0 | -0.023 | -0.001 | -0.004 | 0.087 | 0.019 | 0.075 | 0.219 | 0.271 | 0.063 |
| | Moderate | -0.078 | 0.0 | 0.01 | -0.003 | -0.033 | 0.003 | 0.126 | 0.106 | 0.173 | -0.039 | 0.026 |
| | Slightly conservative | -0.054 | 0.0 | 0.006 | -0.003 | 0.073 | 0.022 | 0.116 | 0.134 | 0.223 | 0.133 | 0.065 |
| | Conservative | -0.08 | -0.019 | -0.018 | 0.008 | 0.18 | 0.145 | 0.031 | 0.172 | 0.148 | 0.15 | 0.072 |
| | Extremely conservative | 0.031 | -0.004 | 0.011 | 0.04 | 0.202 | 0.131 | 0.076 | 0.101 | 0.023 | 0.093 | 0.07 |
| Party | Strong democrat | -0.021 | -0.002 | -0.014 | -0.015 | 0.005 | 0.074 | 0.014 | 0.023 | 0.096 | 0.326 | 0.049 |
| | Weak democrat | -0.012 | -0.004 | -0.001 | 0.014 | 0.019 | 0.102 | 0.03 | 0.085 | 0.197 | 0.145 | 0.057 |
| | Democrat-leaning indep. | -0.045 | 0.0 | -0.01 | -0.014 | 0.029 | 0.073 | 0.015 | 0.084 | 0.205 | 0.172 | 0.051 |
| | Independent | -0.09 | -0.035 | -0.002 | 0.001 | 0.012 | 0.032 | 0.055 | 0.084 | 0.116 | 0.11 | 0.028 |
| | Republican-leaning indep. | -0.049 | -0.032 | 0.037 | -0.004 | 0.047 | 0.013 | 0.253 | 0.076 | 0.051 | 0.108 | 0.05 |
| | Weak republican | 0.0 | -0.018 | 0.018 | -0.013 | 0.041 | 0.032 | 0.077 | 0.098 | 0.151 | 0.14 | 0.053 |
| | Strong republican | -0.028 | -0.032 | -0.001 | -0.007 | 0.067 | 0.081 | -0.01 | 0.018 | 0.08 | 0.221 | 0.039 |
| Church | Attend church | -0.042 | -0.012 | -0.001 | 0.001 | 0.031 | 0.035 | 0.078 | 0.044 | 0.116 | 0.18 | 0.043 |
| | Does not attend church | -0.032 | -0.022 | -0.0 | 0.001 | 0.051 | 0.065 | 0.058 | 0.063 | 0.135 | 0.193 | 0.051 |
| Gender | Man | -0.036 | -0.023 | -0.0 | 0.001 | 0.044 | 0.04 | 0.075 | 0.059 | 0.123 | 0.133 | 0.042 |
| | Woman | -0.037 | -0.017 | 0.0 | 0.001 | 0.042 | 0.056 | 0.061 | 0.044 | 0.128 | 0.23 | 0.051 |
| Political Interest | Very | -0.016 | -0.015 | 0.0 | -0.002 | 0.049 | 0.019 | 0.046 | 0.041 | 0.115 | 0.216 | 0.045 |
| | Somewhat | -0.009 | -0.016 | -0.0 | 0.003 | 0.037 | 0.073 | 0.058 | 0.05 | 0.148 | 0.192 | 0.054 |
| | Not very | -0.025 | -0.027 | -0.002 | 0.002 | 0.052 | 0.043 | 0.116 | 0.082 | 0.124 | 0.15 | 0.051 |
| | Not at all | -0.188 | -0.027 | -0.004 | 0.007 | -0.006 | 0.075 | 0.062 | 0.065 | 0.098 | 0.157 | 0.024 |

Table 17: Difference in JS-divergence ($\Delta$) between **Reverse-coded** and Replicate conditions on stratified subgroups using **Llama-70B**. The color gradient indicates the magnitude of the change: Darker green signifies greater improvement in alignment ($\Delta < 0$), while darker red signifies greater worsening of alignment ($\Delta > 0$).

| | | Climate Change | Refugee Allowing | Race Diversity | Gay Marriage | Income Inequality | Gender Role | Average |
|---|---|---|---|---|---|---|---|---|
| Race | White | -0.06 | 0.005 | 0.039 | 0.081 | -0.008 | 0.768 | 0.107 |
| | Black | -0.094 | 0.082 | 0.031 | 0.042 | 0.023 | 0.778 | 0.138 |
| | Asian | -0.09 | -0.003 | 0.012 | 0.044 | 0.004 | 0.712 | 0.098 |
| | Native American | -0.075 | -0.008 | 0.009 | 0.09 | 0.014 | 0.668 | 0.1 |
| | Hispanic | -0.111 | 0.029 | 0.026 | 0.013 | -0.01 | 0.692 | 0.086 |
| Discuss Politics | Like to discuss politics | -0.076 | 0.007 | 0.036 | 0.073 | -0.001 | 0.759 | 0.105 |
| | Never discuss politics | 0.037 | 0.021 | 0.067 | 0.055 | -0.031 | 0.775 | 0.153 |
| Ideology | Extremely liberal | 0.057 | 0.0 | 0.0 | 0.011 | 0.729 | 0.436 | 0.198 |
| | Liberal | 0.009 | 0.0 | 0.0 | -0.008 | 0.463 | 0.729 | 0.188 |
| | Slightly liberal | -0.267 | 0.019 | 0.0 | -0.021 | 0.192 | 0.785 | 0.114 |
| | Moderate | -0.234 | 0.048 | 0.0 | 0.022 | 0.008 | 0.78 | 0.107 |
| | Slightly conservative | -0.056 | 0.003 | 0.08 | 0.044 | 0.207 | 0.786 | 0.152 |
| | Conservative | -0.01 | 0.013 | 0.132 | 0.154 | 0.535 | 0.717 | 0.197 |
| | Extremely conservative | -0.095 | -0.087 | -0.012 | 0.14 | 0.615 | 0.24 | 0.104 |
| Party | Strong democrat | -0.129 | 0.019 | 0.007 | 0.035 | 0.275 | 0.769 | 0.152 |
| | Weak democrat | -0.093 | 0.018 | 0.022 | -0.004 | 0.133 | 0.779 | 0.13 |
| | Democrat-leaning indep. | -0.129 | 0.021 | 0.003 | 0.018 | 0.246 | 0.808 | 0.142 |
| | Independent | 0.018 | 0.042 | 0.042 | 0.069 | -0.02 | 0.759 | 0.127 |
| | Republican-leaning indep. | -0.07 | 0.07 | 0.068 | 0.049 | 0.198 | 0.791 | 0.137 |
| | Weak republican | -0.026 | 0.027 | 0.072 | 0.034 | 0.067 | 0.797 | 0.103 |
| | Strong republican | 0.011 | 0.0 | 0.074 | 0.161 | 0.308 | 0.664 | 0.145 |
| Church | Attend church | -0.064 | 0.015 | 0.043 | 0.112 | -0.013 | 0.748 | 0.112 |
| | Does not attend church | -0.066 | -0.0 | 0.031 | 0.043 | 0.016 | 0.764 | 0.106 |
| Gender | Man | -0.07 | 0.012 | 0.038 | 0.085 | -0.006 | 0.731 | 0.101 |
| | Woman | -0.063 | 0.005 | 0.035 | 0.062 | 0.001 | 0.769 | 0.112 |
| Political Interest | Very | -0.121 | -0.003 | 0.023 | 0.09 | 0.004 | 0.725 | 0.093 |
| | Somewhat | -0.082 | 0.011 | 0.027 | 0.078 | -0.0 | 0.766 | 0.105 |
| | Not very | -0.05 | 0.008 | 0.078 | 0.055 | -0.02 | 0.797 | 0.12 |
| | Not at all | 0.019 | 0.015 | 0.035 | 0.028 | -0.016 | 0.731 | 0.12 |

Table 18: Difference in JS-divergence (Δ) between **Priming** and Replicate conditions on stratified subgroups using **Llama-70B**. The color gradient indicates the magnitude of the change: Darker green signifies greater improvement in alignment (Δ < 0), while darker red signifies greater worsening of alignment (Δ > 0).

| | | Gay Marriage | Drug Addiction | Income Inequality | Health Insurance | Race Diversity | Refugee Allowing | Gun Regulation | Climate Change | Gender Role | Current Economy | Average |
|---|---|---|---|---|---|---|---|---|---|---|---|---|
| Race | White | -0.004 | 0.002 | -0.005 | 0.003 | 0.007 | 0.013 | 0.025 | 0.094 | 0.121 | 0.146 | 0.04 |
| | Black | -0.005 | 0.0 | 0.014 | 0.018 | 0.025 | -0.003 | -0.003 | 0.057 | 0.178 | 0.107 | 0.039 |
| | Asian | 0.009 | 0.0 | 0.008 | 0.005 | 0.002 | 0.031 | 0.019 | 0.031 | 0.098 | 0.074 | 0.028 |
| | Native American | 0.01 | 0.007 | 0.012 | -0.001 | -0.021 | 0.021 | 0.026 | -0.006 | 0.07 | 0.132 | 0.025 |
| | Hispanic | -0.043 | 0.0 | -0.008 | -0.001 | 0.026 | -0.008 | 0.007 | 0.097 | 0.142 | 0.104 | 0.032 |
| Discuss Politics | Like to discuss politics | -0.004 | 0.002 | 0.0 | 0.006 | 0.001 | 0.014 | 0.024 | 0.069 | 0.122 | 0.136 | 0.037 |
| | Never discuss politics | -0.016 | 0.0 | -0.017 | -0.001 | 0.033 | -0.013 | 0.068 | 0.068 | 0.082 | 0.136 | 0.034 |
| Ideology | Extremely liberal | 0.011 | 0.0 | 0.0 | 0.0 | 0.0 | 0.0 | 0.007 | 0.042 | -0.266 | 0.018 | -0.019 |
| | Liberal | -0.019 | 0.0 | 0.006 | 0.011 | 0.0 | 0.0 | -0.006 | 0.116 | 0.05 | 0.11 | 0.027 |
| | Slightly liberal | 0.0 | 0.0 | 0.03 | 0.008 | 0.0 | -0.002 | -0.026 | 0.074 | 0.102 | 0.127 | 0.031 |
| | Moderate | 0.001 | 0.0 | 0.004 | 0.008 | 0.0 | -0.034 | 0.063 | 0.079 | 0.102 | 0.194 | 0.042 |
| | Slightly conservative | 0.004 | 0.0 | -0.002 | 0.004 | 0.043 | 0.007 | 0.052 | 0.066 | 0.067 | 0.248 | 0.049 |
| | Conservative | 0.021 | 0.0 | -0.003 | 0.003 | 0.031 | 0.099 | 0.008 | 0.069 | 0.103 | 0.201 | 0.053 |
| | Extremely conservative | 0.037 | 0.023 | 0.069 | 0.025 | -0.101 | 0.125 | -0.018 | -0.009 | -0.047 | 0.226 | 0.033 |
| Party | Strong democrat | -0.003 | 0.0 | -0.008 | -0.014 | -0.005 | 0.016 | -0.0 | 0.031 | 0.105 | 0.064 | 0.019 |
| | Weak democrat | -0.014 | 0.0 | 0.009 | 0.011 | 0.014 | 0.013 | -0.001 | 0.095 | 0.152 | 0.082 | 0.036 |
| | Democrat-leaning indep. | -0.002 | 0.0 | -0.0 | -0.008 | 0.003 | -0.005 | 0.004 | 0.139 | 0.109 | 0.115 | 0.035 |
| | Independent | -0.008 | 0.0 | 0.004 | 0.003 | 0.035 | 0.004 | 0.004 | 0.178 | 0.152 | 0.129 | 0.05 |
| | Republican-leaning indep. | -0.019 | 0.0 | -0.008 | 0.011 | 0.051 | 0.009 | 0.162 | 0.194 | 0.154 | 0.14 | 0.069 |
| | Weak republican | -0.0 | 0.0 | -0.02 | -0.001 | 0.046 | 0.007 | 0.022 | 0.141 | 0.129 | 0.157 | 0.048 |
| | Strong republican | -0.017 | 0.007 | 0.001 | -0.004 | -0.014 | 0.01 | -0.031 | 0.04 | 0.116 | 0.176 | 0.028 |
| Church | Attend church | -0.005 | 0.0 | -0.013 | 0.002 | 0.011 | 0.006 | 0.034 | 0.072 | 0.12 | 0.129 | 0.035 |
| | Does not attend church | 0.0 | 0.003 | 0.01 | 0.005 | 0.004 | 0.017 | 0.017 | 0.089 | 0.13 | 0.136 | 0.041 |
| Gender | Man | -0.002 | 0.003 | 0.0 | -0.001 | 0.016 | 0.008 | 0.026 | 0.099 | 0.125 | 0.147 | 0.042 |
| | Woman | -0.005 | 0.001 | -0.002 | 0.008 | 0.001 | 0.017 | 0.026 | 0.066 | 0.107 | 0.124 | 0.034 |
| Political Interest | Very | 0.004 | 0.0 | 0.002 | 0.007 | 0.015 | 0.023 | 0.027 | 0.078 | 0.107 | 0.101 | 0.036 |
| | Somewhat | -0.004 | 0.0 | 0.0 | 0.005 | 0.019 | 0.01 | 0.022 | 0.073 | 0.107 | 0.132 | 0.036 |
| | Not very | -0.002 | 0.004 | -0.008 | 0.004 | 0.006 | 0.004 | 0.039 | 0.072 | 0.128 | 0.269 | 0.052 |
| | Not at all | -0.02 | 0.012 | -0.011 | -0.001 | -0.043 | 0.004 | 0.023 | 0.037 | 0.167 | 0.246 | 0.041 |

Table 19: Difference in JS-divergence (Δ) between **Preamble** and Replicate conditions on stratified subgroups using **Llama-70B**. The color gradient indicates the magnitude of the change: Darker green signifies greater improvement in alignment (Δ < 0), while darker red signifies greater worsening of alignment (Δ > 0).

| | | Refugee Allowing | Drug Addiction | Income Inequality | Health Insurance | Race Diversity | Climate Change | Gun Regulation | Gay Marriage | Current Economy | Gender Role | Average |
|---|---|---|---|---|---|---|---|---|---|---|---|---|
| Race | White | -0.029 | -0.019 | -0.001 | 0.005 | 0.06 | 0.052 | 0.074 | 0.053 | 0.127 | 0.203 | 0.052 |
| | Black | -0.048 | -0.023 | 0.002 | -0.012 | -0.026 | 0.068 | 0.005 | 0.005 | 0.074 | 0.139 | 0.018 |
| | Asian | -0.048 | -0.026 | 0.0 | -0.002 | 0.008 | 0.038 | 0.035 | 0.069 | 0.128 | 0.168 | 0.037 |
| | Native American | -0.044 | -0.004 | -0.013 | -0.003 | 0.038 | 0.038 | 0.075 | 0.069 | 0.202 | 0.109 | 0.047 |
| | Hispanic | -0.032 | -0.017 | 0.014 | -0.003 | 0.004 | 0.048 | 0.019 | -0.028 | 0.069 | 0.223 | 0.03 |
| Discuss Politics | Like to discuss politics | -0.032 | -0.02 | -0.001 | 0.004 | 0.048 | 0.049 | 0.068 | 0.056 | 0.138 | 0.19 | 0.05 |
| | Never discuss politics | -0.11 | -0.023 | 0.002 | -0.001 | -0.031 | 0.029 | 0.111 | 0.031 | 0.14 | 0.123 | 0.027 |
| Ideology | Extremely liberal | 0.0 | 0.0 | 0.0 | 0.0 | 0.0 | 0.024 | 0.007 | 0.004 | 0.008 | 0.423 | 0.047 |
| | Liberal | 0.0 | 0.0 | 0.0 | 0.011 | 0.0 | 0.113 | -0.008 | 0.022 | 0.158 | 0.612 | 0.091 |
| | Slightly liberal | -0.009 | 0.0 | -0.023 | -0.001 | -0.004 | 0.087 | 0.019 | 0.075 | 0.219 | 0.271 | 0.063 |
| | Moderate | -0.078 | 0.0 | 0.01 | -0.003 | -0.033 | 0.003 | 0.126 | 0.106 | 0.173 | -0.039 | 0.026 |
| | Slightly conservative | -0.054 | 0.0 | 0.006 | -0.003 | 0.073 | 0.022 | 0.116 | 0.134 | 0.223 | 0.133 | 0.065 |
| | Conservative | -0.08 | -0.019 | -0.018 | 0.008 | 0.18 | 0.145 | 0.031 | 0.172 | 0.148 | 0.15 | 0.072 |
| | Extremely conservative | 0.031 | -0.004 | 0.011 | 0.04 | 0.202 | 0.131 | 0.076 | 0.101 | 0.023 | 0.093 | 0.07 |
| Party | Strong democrat | -0.021 | -0.002 | -0.014 | -0.015 | 0.005 | 0.074 | 0.014 | 0.023 | 0.096 | 0.326 | 0.049 |
| | Weak democrat | -0.012 | -0.004 | -0.001 | 0.014 | 0.019 | 0.102 | 0.03 | 0.085 | 0.197 | 0.145 | 0.057 |
| | Democrat-leaning indep. | -0.045 | 0.0 | -0.01 | -0.014 | 0.029 | 0.073 | 0.015 | 0.084 | 0.205 | 0.172 | 0.051 |
| | Independent | -0.09 | -0.035 | -0.002 | 0.001 | 0.012 | 0.032 | 0.055 | 0.084 | 0.116 | 0.11 | 0.028 |
| | Republican-leaning indep. | -0.049 | -0.032 | 0.037 | -0.004 | 0.047 | 0.013 | 0.253 | 0.076 | 0.051 | 0.108 | 0.05 |
| | Weak republican | 0.0 | -0.018 | 0.018 | -0.013 | 0.041 | 0.032 | 0.077 | 0.098 | 0.151 | 0.14 | 0.053 |
| | Strong republican | -0.028 | -0.032 | -0.001 | -0.007 | 0.067 | 0.081 | -0.01 | 0.018 | 0.08 | 0.221 | 0.039 |
| Church | Attend church | -0.042 | -0.012 | -0.001 | 0.001 | 0.031 | 0.035 | 0.078 | 0.044 | 0.116 | 0.18 | 0.043 |
| | Does not attend church | -0.032 | -0.022 | -0.0 | 0.001 | 0.051 | 0.065 | 0.058 | 0.063 | 0.135 | 0.193 | 0.051 |
| Gender | Man | -0.036 | -0.023 | -0.0 | 0.001 | 0.044 | 0.04 | 0.075 | 0.059 | 0.123 | 0.133 | 0.042 |
| | Woman | -0.037 | -0.017 | 0.0 | 0.001 | 0.042 | 0.056 | 0.061 | 0.044 | 0.128 | 0.23 | 0.051 |
| Political Interest | Very | -0.016 | -0.015 | 0.0 | -0.002 | 0.049 | 0.019 | 0.046 | 0.041 | 0.115 | 0.216 | 0.045 |
| | Somewhat | -0.009 | -0.016 | -0.0 | 0.003 | 0.037 | 0.073 | 0.058 | 0.05 | 0.148 | 0.192 | 0.054 |
| | Not very | -0.025 | -0.027 | -0.002 | 0.002 | 0.052 | 0.043 | 0.116 | 0.082 | 0.124 | 0.15 | 0.051 |
| | Not at all | -0.188 | -0.027 | -0.004 | 0.007 | -0.006 | 0.075 | 0.062 | 0.065 | 0.098 | 0.157 | 0.024 |

Table 20: Difference in JS-divergence (Δ) between **Reformulated** and Replicate conditions on stratified subgroups using **GPT-4.1-mini**. The color gradient indicates the magnitude of the change: Darker green signifies greater improvement in alignment (Δ < 0), while darker red signifies greater worsening of alignment (Δ > 0).

| | | Refugee Allowing | Race Diversity | Gender Role | Current Economy | Climate Change | Drug Addiction | Gay Marriage | Gun Regulation | Health Insurance | Income Inequality | Average |
|---|---|---|---|---|---|---|---|---|---|---|---|---|
| Race | White | -0.08 | -0.071 | -0.057 | -0.07 | -0.046 | -0.02 | -0.006 | 0.009 | 0.024 | 0.06 | -0.026 |
| | Black | -0.137 | -0.076 | -0.097 | -0.018 | 0.024 | -0.027 | -0.023 | 0.018 | 0.026 | 0.068 | -0.024 |
| | Asian | -0.093 | -0.028 | -0.066 | -0.045 | -0.007 | -0.029 | -0.014 | 0.015 | 0.007 | 0.059 | -0.02 |
| | Native American | -0.073 | -0.046 | -0.084 | -0.047 | 0.055 | -0.009 | -0.02 | 0.051 | 0.061 | 0.089 | -0.002 |
| | Hispanic | -0.179 | -0.028 | -0.094 | -0.055 | -0.038 | -0.039 | 0.007 | 0.005 | 0.034 | 0.049 | -0.034 |
| Discuss Politics | Like to discuss politics | -0.081 | -0.054 | -0.074 | -0.056 | -0.029 | -0.021 | -0.008 | 0.017 | 0.034 | 0.057 | -0.021 |
| | Never discuss politics | -0.204 | -0.156 | -0.082 | -0.053 | -0.059 | -0.029 | -0.035 | -0.026 | 0.023 | 0.077 | -0.054 |
| Ideology | Extremely liberal | -0.034 | -0.008 | 0.29 | -0.124 | -0.012 | 0.0 | 0.0 | 0.049 | 0.013 | 0.013 | 0.019 |
| | Liberal | -0.054 | -0.008 | -0.001 | -0.07 | 0.03 | 0.0 | -0.008 | 0.028 | 0.012 | 0.032 | -0.004 |
| | Slightly liberal | -0.033 | -0.014 | -0.103 | 0.024 | 0.051 | -0.012 | -0.013 | 0.033 | 0.015 | 0.066 | 0.001 |
| | Moderate | -0.127 | -0.056 | -0.096 | -0.025 | -0.014 | -0.011 | -0.016 | 0.007 | 0.061 | 0.044 | -0.023 |
| | Slightly conservative | -0.151 | -0.083 | -0.074 | 0.048 | -0.097 | -0.041 | -0.03 | -0.0 | 0.044 | 0.053 | -0.033 |
| | Conservative | -0.091 | -0.144 | -0.059 | -0.093 | -0.088 | -0.032 | -0.003 | 0.012 | 0.023 | 0.071 | -0.04 |
| | Extremely conservative | -0.004 | -0.101 | -0.039 | -0.231 | -0.175 | 0.044 | 0.045 | 0.161 | 0.045 | 0.069 | -0.019 |
| Party | Strong democrat | -0.031 | 0.0 | -0.016 | -0.074 | -0.013 | 0.0 | 0.0 | 0.022 | 0.0 | 0.012 | -0.01 |
| | Weak democrat | -0.071 | -0.032 | -0.067 | 0.072 | 0.051 | 0.0 | 0.027 | -0.01 | 0.033 | 0.044 | 0.005 |
| | Democrat-leaning indep. | -0.025 | 0.0 | -0.022 | 0.027 | 0.006 | 0.0 | 0.0 | -0.008 | 0.0 | 0.007 | -0.001 |
| | Independent | -0.156 | -0.103 | -0.129 | 0.05 | -0.077 | -0.032 | 0.012 | -0.023 | 0.027 | 0.083 | -0.035 |
| | Republican-leaning indep. | -0.176 | -0.135 | -0.141 | 0.087 | -0.085 | -0.037 | -0.008 | 0.034 | 0.022 | -0.057 | -0.049 |
| | Weak republican | -0.106 | -0.105 | -0.07 | 0.007 | -0.097 | -0.024 | -0.007 | -0.006 | 0.001 | 0.036 | -0.037 |
| | Strong republican | -0.016 | -0.117 | -0.003 | -0.194 | 0.009 | 0.017 | -0.034 | 0.229 | 0.071 | 0.153 | 0.012 |
| Church | Attend church | -0.102 | -0.079 | -0.073 | -0.068 | -0.05 | -0.024 | -0.004 | 0.006 | 0.025 | 0.045 | -0.032 |
| | Does not attend church | -0.073 | -0.057 | -0.061 | -0.049 | -0.011 | -0.02 | -0.014 | 0.016 | 0.023 | 0.078 | -0.017 |
| Gender | Man | -0.072 | -0.085 | -0.082 | -0.088 | -0.056 | -0.021 | -0.005 | 0.01 | 0.03 | 0.055 | -0.031 |
| | Woman | -0.098 | -0.053 | -0.052 | -0.038 | -0.011 | -0.022 | -0.012 | 0.017 | 0.02 | 0.065 | -0.018 |
| Political Interest | Very | -0.045 | -0.047 | -0.053 | -0.08 | -0.056 | -0.005 | -0.0 | 0.031 | 0.02 | 0.05 | -0.019 |
| | Somewhat | -0.067 | -0.044 | -0.066 | -0.019 | -0.026 | -0.019 | -0.007 | 0.017 | 0.036 | 0.089 | -0.011 |
| | Not very | -0.142 | -0.109 | -0.093 | -0.056 | -0.019 | -0.027 | -0.021 | -0.026 | -0.003 | 0.033 | -0.046 |
| | Not at all | -0.068 | -0.133 | -0.107 | -0.131 | 0.02 | -0.03 | 0.0 | -0.019 | 0.034 | 0.002 | -0.043 |

Table 21: Difference in JS-divergence (Δ) between **Reverse-coded** and Replicate conditions on stratified subgroups using **GPT-4.1-mini**. The color gradient indicates the magnitude of the change: Darker green signifies greater improvement in alignment (Δ < 0), while darker red signifies greater worsening of alignment (Δ > 0).

| | | Race Diversity | Refugee Allowing | Income Inequality | Gay Marriage | Climate Change | Gender Role | Average |
|---|---|---|---|---|---|---|---|---|
| Race | White | -0.13 | -0.056 | -0.038 | 0.057 | 0.278 | 0.802 | 0.144 |
| | Black | -0.114 | -0.127 | -0.043 | 0.03 | 0.275 | 0.664 | 0.131 |
| | Asian | -0.204 | -0.074 | -0.036 | 0.058 | 0.316 | 0.711 | 0.125 |
| | Native American | -0.156 | -0.083 | -0.002 | 0.032 | 0.306 | 0.662 | 0.124 |
| | Hispanic | -0.203 | -0.141 | -0.101 | 0.058 | 0.278 | 0.689 | 0.103 |
| Discuss Politics | Like to discuss politics | -0.135 | -0.065 | -0.036 | 0.054 | 0.277 | 0.767 | 0.138 |
| | Never discuss politics | -0.185 | -0.1 | -0.083 | 0.032 | 0.358 | 0.732 | 0.138 |
| Ideology | Extremely liberal | -0.012 | -0.011 | 0.176 | 0.0 | 0.082 | 0.713 | 0.147 |
| | Liberal | 0.0 | -0.019 | 0.104 | -0.008 | 0.158 | 0.673 | 0.139 |
| | Slightly liberal | -0.022 | -0.016 | 0.072 | 0.008 | 0.206 | 0.706 | 0.148 |
| | Moderate | -0.16 | -0.127 | -0.049 | 0.034 | 0.32 | 0.745 | 0.134 |
| | Slightly conservative | -0.107 | -0.202 | -0.037 | 0.075 | 0.451 | 0.792 | 0.164 |
| | Conservative | -0.263 | -0.04 | 0.183 | 0.159 | 0.467 | 0.855 | 0.21 |
| | Extremely conservative | -0.335 | -0.023 | 0.311 | 0.056 | 0.175 | 0.81 | 0.154 |
| Party | Strong democrat | -0.02 | -0.026 | 0.031 | -0.035 | 0.152 | 0.67 | 0.129 |
| | Weak democrat | -0.087 | -0.09 | -0.014 | 0.032 | 0.244 | 0.696 | 0.131 |
| | Democrat-leaning indep. | -0.036 | -0.025 | 0.038 | -0.016 | 0.204 | 0.701 | 0.137 |
| | Independent | -0.113 | -0.148 | -0.09 | 0.056 | 0.361 | 0.673 | 0.133 |
| | Republican-leaning indep. | -0.203 | -0.114 | 0.062 | 0.073 | 0.416 | 0.74 | 0.165 |
| | Weak republican | -0.173 | -0.072 | -0.094 | 0.088 | 0.427 | 0.822 | 0.163 |
| | Strong republican | -0.31 | 0.048 | 0.522 | 0.082 | 0.415 | 0.87 | 0.254 |
| Church | Attend church | -0.131 | -0.065 | -0.061 | 0.087 | 0.335 | 0.786 | 0.154 |
| | Does not attend church | -0.142 | -0.057 | -0.016 | 0.029 | 0.242 | 0.757 | 0.13 |
| Gender | Man | -0.16 | -0.057 | -0.028 | 0.064 | 0.288 | 0.763 | 0.138 |
| | Woman | -0.121 | -0.071 | -0.046 | 0.045 | 0.277 | 0.775 | 0.139 |
| Political Interest | Very | -0.089 | -0.008 | -0.024 | 0.047 | 0.158 | 0.763 | 0.128 |
| | Somewhat | -0.134 | -0.069 | -0.031 | 0.063 | 0.275 | 0.777 | 0.142 |
| | Not very | -0.177 | -0.129 | -0.078 | 0.048 | 0.395 | 0.762 | 0.135 |
| | Not at all | -0.238 | -0.089 | -0.06 | 0.015 | 0.414 | 0.697 | 0.128 |

Table 22: Difference in JS-divergence (Δ) between **Priming** and Replicate conditions on stratified subgroups using **GPT-4.1-mini**. The color gradient indicates the magnitude of the change: Darker green signifies greater improvement in alignment (Δ < 0), while darker red signifies greater worsening of alignment (Δ > 0).

| | | Gay Marriage | Refugee Allowing | Drug Addiction | Gun Regulation | Race Diversity | Health Insurance | Current Economy | Income Inequality | Gender Role | Climate Change | Average |
|---|---|---|---|---|---|---|---|---|---|---|---|---|
| Race | White | -0.015 | -0.0 | 0.008 | 0.013 | 0.018 | 0.038 | 0.052 | 0.05 | 0.11 | 0.156 | 0.043 |
| | Black | -0.005 | -0.028 | 0.006 | 0.005 | 0.0 | 0.012 | 0.023 | -0.002 | 0.093 | 0.152 | 0.026 |
| | Asian | -0.003 | -0.015 | 0.0 | 0.006 | 0.0 | 0.048 | 0.023 | 0.043 | 0.1 | 0.169 | 0.037 |
| | Native American | -0.022 | 0.029 | 0.0 | -0.001 | 0.0 | 0.039 | 0.015 | 0.006 | 0.073 | 0.108 | 0.025 |
| | Hispanic | 0.005 | -0.066 | 0.0 | 0.024 | 0.0 | 0.048 | 0.069 | 0.032 | 0.066 | 0.181 | 0.036 |
| Discuss Politics | Like to discuss politics | -0.012 | -0.002 | 0.007 | 0.01 | 0.007 | 0.031 | 0.055 | 0.046 | 0.1 | 0.147 | 0.039 |
| | Never discuss politics | -0.016 | 0.037 | 0.006 | 0.005 | 0.044 | 0.025 | 0.043 | 0.035 | 0.108 | 0.246 | 0.053 |
| Ideology | Extremely liberal | 0.0 | 0.009 | 0.0 | 0.03 | 0.0 | 0.013 | 0.096 | 0.036 | 0.049 | 0.082 | 0.031 |
| | Liberal | -0.003 | 0.008 | 0.0 | -0.007 | 0.0 | 0.021 | 0.02 | 0.056 | 0.004 | 0.109 | 0.021 |
| | Slightly liberal | -0.018 | 0.004 | 0.0 | 0.015 | 0.0 | 0.07 | 0.035 | 0.113 | 0.023 | 0.056 | 0.03 |
| | Moderate | -0.001 | 0.017 | 0.0 | 0.01 | 0.0 | 0.09 | 0.081 | 0.079 | 0.066 | 0.139 | 0.048 |
| | Slightly conservative | -0.017 | 0.005 | 0.0 | 0.032 | 0.02 | 0.115 | 0.023 | 0.142 | 0.118 | 0.29 | 0.073 |
| | Conservative | -0.001 | 0.054 | 0.003 | -0.01 | 0.01 | 0.034 | 0.033 | 0.091 | 0.198 | 0.361 | 0.079 |
| | Extremely conservative | -0.072 | -0.036 | 0.052 | -0.01 | 0.103 | 0.001 | 0.042 | 0.018 | 0.221 | 0.395 | 0.071 |
| Party | Strong democrat | -0.003 | -0.003 | 0.0 | -0.009 | 0.0 | 0.0 | 0.048 | 0.002 | 0.014 | 0.113 | 0.016 |
| | Weak democrat | 0.027 | 0.023 | 0.0 | 0.011 | 0.0 | 0.047 | 0.042 | 0.058 | 0.045 | 0.095 | 0.035 |
| | Democrat-leaning indep. | 0.0 | 0.005 | 0.0 | 0.008 | 0.0 | 0.0 | 0.04 | -0.005 | 0.008 | 0.052 | 0.011 |
| | Independent | 0.014 | 0.025 | 0.0 | 0.028 | 0.012 | 0.062 | 0.077 | 0.02 | 0.033 | 0.2 | 0.047 |
| | Republican-leaning indep. | -0.005 | 0.028 | 0.0 | 0.002 | 0.008 | 0.087 | 0.011 | 0.103 | 0.08 | 0.287 | 0.06 |
| | Weak republican | -0.004 | 0.042 | 0.005 | 0.012 | 0.035 | 0.041 | 0.02 | 0.07 | 0.093 | 0.272 | 0.059 |
| | Strong republican | -0.095 | -0.012 | 0.021 | -0.037 | 0.033 | 0.029 | 0.038 | 0.068 | 0.237 | 0.339 | 0.062 |
| Church | Attend church | -0.004 | 0.006 | 0.005 | 0.009 | 0.005 | 0.054 | 0.052 | 0.062 | 0.12 | 0.183 | 0.049 |
| | Does not attend church | -0.023 | -0.011 | 0.008 | 0.01 | 0.019 | 0.017 | 0.044 | 0.025 | 0.096 | 0.129 | 0.032 |
| Gender | Man | -0.009 | 0.004 | 0.009 | 0.017 | 0.013 | 0.036 | 0.049 | 0.052 | 0.11 | 0.155 | 0.044 |
| | Woman | -0.014 | -0.01 | 0.005 | 0.006 | 0.013 | 0.034 | 0.043 | 0.031 | 0.102 | 0.149 | 0.036 |
| Political Interest | Very | -0.027 | -0.02 | 0.009 | 0.007 | 0.0 | 0.016 | 0.012 | 0.04 | 0.111 | 0.109 | 0.026 |
| | Somewhat | -0.008 | -0.007 | 0.007 | 0.01 | 0.009 | 0.026 | 0.052 | 0.044 | 0.095 | 0.128 | 0.036 |
| | Not very | -0.005 | 0.002 | 0.0 | -0.002 | 0.032 | 0.048 | 0.058 | 0.031 | 0.095 | 0.189 | 0.045 |
| | Not at all | 0.0 | 0.042 | 0.0 | 0.028 | 0.037 | 0.068 | 0.036 | 0.028 | 0.095 | 0.245 | 0.058 |

Table 23: Difference in JS-divergence (Δ) between **Preamble** and Replicate conditions on stratified subgroups using **GPT-4.1-mini**. The color gradient indicates the magnitude of the change: Darker green signifies greater improvement in alignment (Δ < 0), while darker red signifies greater worsening of alignment (Δ > 0).

| | | Race Diversity | Current Economy | Drug Addiction | Gun Regulation | Gay Marriage | Climate Change | Gender Role | Income Inequality | Refugee Allowing | Health Insurance | Average |
|---|---|---|---|---|---|---|---|---|---|---|---|---|
| Race | White | -0.002 | -0.008 | 0.008 | 0.012 | 0.024 | 0.023 | 0.067 | 0.058 | 0.141 | 0.17 | 0.049 |
| | Black | -0.008 | -0.027 | 0.006 | -0.003 | -0.009 | 0.017 | 0.03 | 0.002 | 0.094 | 0.102 | 0.02 |
| | Asian | -0.027 | -0.044 | 0.0 | 0.004 | 0.015 | 0.021 | 0.077 | 0.019 | 0.116 | 0.13 | 0.031 |
| | Native American | -0.016 | -0.016 | 0.0 | -0.004 | -0.032 | 0.029 | 0.037 | 0.021 | 0.133 | 0.165 | 0.032 |
| | Hispanic | -0.047 | 0.005 | 0.0 | 0.024 | 0.005 | 0.041 | 0.028 | 0.014 | 0.085 | 0.225 | 0.038 |
| Discuss Politics | Like to discuss politics | -0.006 | -0.017 | 0.007 | 0.01 | 0.021 | 0.022 | 0.059 | 0.046 | 0.13 | 0.166 | 0.044 |
| | Never discuss politics | -0.016 | 0.007 | 0.006 | 0.003 | 0.004 | 0.039 | 0.072 | 0.072 | 0.176 | 0.181 | 0.051 |
| Ideology | Extremely liberal | 0.0 | -0.124 | 0.0 | 0.062 | 0.0 | 0.041 | 0.039 | 0.036 | 0.009 | 0.013 | 0.008 |
| | Liberal | 0.0 | -0.055 | 0.0 | 0.006 | 0.0 | 0.033 | 0.004 | 0.037 | 0.013 | 0.021 | 0.006 |
| | Slightly liberal | 0.0 | 0.028 | 0.0 | -0.0 | 0.012 | -0.006 | 0.023 | 0.058 | 0.071 | 0.094 | 0.028 |
| | Moderate | 0.0 | 0.002 | 0.0 | -0.002 | 0.043 | 0.015 | 0.066 | 0.098 | 0.12 | 0.197 | 0.054 |
| | Slightly conservative | 0.02 | 0.017 | 0.0 | 0.031 | 0.033 | 0.086 | 0.096 | 0.117 | 0.132 | 0.308 | 0.084 |
| | Conservative | -0.026 | 0.004 | 0.003 | 0.021 | 0.042 | 0.066 | 0.101 | 0.107 | 0.318 | 0.366 | 0.1 |
| | Extremely conservative | -0.049 | 0.023 | 0.052 | 0.039 | 0.03 | 0.05 | 0.04 | 0.019 | 0.211 | 0.241 | 0.065 |
| Party | Strong democrat | 0.0 | -0.065 | 0.0 | 0.023 | 0.0 | 0.027 | 0.011 | 0.009 | 0.0 | 0.0 | 0.001 |
| | Weak democrat | -0.007 | 0.013 | 0.0 | 0.004 | 0.04 | -0.003 | 0.035 | 0.046 | 0.055 | 0.083 | 0.027 |
| | Democrat-leaning indep. | 0.0 | -0.013 | 0.0 | 0.07 | 0.0 | -0.001 | 0.008 | 0.024 | 0.005 | 0.0 | 0.009 |
| | Independent | -0.015 | 0.007 | 0.0 | 0.022 | 0.035 | 0.026 | -0.003 | 0.051 | 0.132 | 0.198 | 0.045 |
| | Republican-leaning indep. | 0.0 | 0.029 | 0.0 | -0.015 | 0.021 | 0.053 | 0.059 | 0.136 | 0.159 | 0.29 | 0.073 |
| | Weak republican | -0.027 | -0.002 | 0.005 | 0.027 | 0.034 | 0.068 | 0.049 | 0.09 | 0.2 | 0.25 | 0.069 |
| | Strong republican | -0.003 | 0.032 | 0.021 | -0.04 | -0.003 | 0.043 | 0.124 | 0.071 | 0.321 | 0.323 | 0.089 |
| Church | Attend church | -0.009 | 0.004 | 0.005 | 0.011 | 0.038 | 0.038 | 0.07 | 0.065 | 0.155 | 0.18 | 0.056 |
| | Does not attend church | -0.004 | -0.023 | 0.008 | 0.008 | 0.001 | 0.01 | 0.058 | 0.029 | 0.12 | 0.138 | 0.035 |
| Gender | Man | -0.014 | -0.008 | 0.009 | 0.012 | 0.025 | 0.011 | 0.05 | 0.043 | 0.149 | 0.171 | 0.045 |
| | Woman | 0.001 | -0.016 | 0.005 | 0.009 | 0.014 | 0.031 | 0.075 | 0.051 | 0.125 | 0.155 | 0.045 |
| Political Interest | Very | -0.019 | -0.001 | 0.009 | 0.009 | 0.011 | 0.01 | 0.052 | 0.031 | 0.084 | 0.116 | 0.03 |
| | Somewhat | -0.011 | 0.025 | 0.007 | 0.007 | 0.02 | 0.021 | 0.058 | 0.044 | 0.127 | 0.149 | 0.045 |
| | Not very | 0.004 | 0.01 | 0.0 | 0.018 | 0.035 | 0.014 | 0.067 | 0.064 | 0.168 | 0.222 | 0.06 |
| | Not at all | 0.015 | 0.037 | 0.0 | 0.045 | 0.03 | 0.045 | 0.054 | 0.077 | 0.241 | 0.27 | 0.081 |

### A.8   All Questions and prompt conditions

**Race diversity V202371**

*Original*: Does the increasing number of people of many different races and ethnic groups in the United States make this country a better place to live, a worse place to live, or does it make no difference?

1. Better  2. Worse  3. Makes no difference

*Reformulated:* How would this respondent describe the impact of the increasing number of people of many different races and ethnic groups on the United States as a place to live?

1. Improves it as a place to live  2. Worsens it as a place to live  3. Makes no difference

*Reverse-coded:* Does limiting the number of people of many different races and ethnic groups in the United States make this country a better place to live, a worse place to live, or does it make no difference?

1. Better  2. Worse  3. Makes no difference

**Gender role V202287**

*Original*: Do you think it is better, worse, or makes no difference for the family as a whole if the man works outside the home and the woman takes care of the home and family?

1. Better  2. Worse  3. Makes no difference

*Reformulated:* How would this respondent assess whether it is better, worse, or makes no difference for the family as a whole if the man works outside the home and the woman takes care of the home and family?

1. Better  2. Worse  3. Makes no difference

*Reverse-coded:* Do you think it is better, worse, or makes no difference for the family as a whole if both the man and the woman share work outside the home and take care of the home and family

1. Better  2. Worse  3. Makes no difference

**Current Economy V201324**

*Original*: What do you think about the state of the economy these days in the United States?

1. Very good  2. Good  3. Neither good nor bad  4. Bad  5. Very bad

*Reformulated:* How would this respondent describe the state of the economy these days in the United States?

1. Very strong  2. Strong  3. Neither strong nor weak  4. Weak  5. Very weak

*Reverse-coded:* Not applicable

**Drug addiction V202348**

*Original*: Do you think the federal government should be doing more about the opioid drug addiction issue, should be doing less, or is it currently doing the right amount?

1. Should be doing more  2. Should be doing less  3. Is doing the right amount

*Reformulated:* How would this respondent assess whether the federal government should be doing more about the opioid drug addiction issue, should be doing less, or is it currently doing the right amount?

1. Should be doing more  2. Should be doing less  3. Is doing the right amount

*Reverse-coded:* Not applicable

**Climate change V202332**

*Original*: How much, if at all, do you think climate change is currently affecting severe weather events or temperature patterns in the United States?

1. Not at all  2. A little  3. A moderate amount  4. A lot  5. A great deal

*Reformulated:* How would this respondent assess how much, if at all, climate change is currently affecting severe weather events or temperature patterns in the United States?

1. Not at all  2. A little  3. A moderate amount  4. A lot  5. A great deal

*Reverse-coded:* How much do you agree with the statement that climate change is unrelated to severe weather events or temperature patterns in the United States?

1. Not at all  2. A little  3. A moderate amount  4. A lot  5. A great deal

**Gay marriage V201416**

*Original*: Which comes closest to your view? You can just tell me the number of your choice.

1. Gay and lesbian couples should be allowed to legally marry.  2. Gay and lesbian couples should be allowed to form civil unions but not legally marry.  3. There should be no legal recognition of gay or lesbian couples' relationship.

*Reformulated:* Which comes closest to this respondent's view?

1. Gay and lesbian couples should be allowed to legally marry.  2. Gay and lesbian couples should be allowed to form civil unions but not legally marry.  3. There should be no legal recognition of gay or lesbian couples' relationship.

*Reverse-coded:* Which one disagrees the most with your view?

1. Gay and lesbian couples should be allowed to legally marry.  2. Gay and lesbian couples should be allowed to form civil unions but not legally marry.  3. There should be no legal recognition of gay or lesbian couples' relationship.

**Refugee allowing V202234**

*Original*: Do you favor, oppose, or neither favor nor oppose allowing refugees who are fleeing war, persecution, or natural disasters in other countries to come to live in the U.S.?

1. Favor  2. Oppose  3. Neither favor nor oppose

*Reformulated:* What is this respondent's position on whether refugees who are fleeing war, persecution, or natural disasters in other countries should be allowed to come to live in the U.S.?

1. Should be allowed  2. Should not be allowed  3. No clear position

*Reverse-coded:* Do you favor, oppose, or neither favor nor oppose prohibiting refugees who are fleeing war, persecution, or natural disasters in other countries from coming to live in the U.S.?

1. Favor  2. Oppose  3. Neither favor nor oppose

**Health insurance V202378**

*Original*: Do you favor an increase, decrease, or no change in government spending to help people pay for health insurance when people cannot pay for it all themselves?

1. Increase  2. Decrease  3. No change

*Reformulated:* How would this respondent assess if there should be an increase, decrease, or no change in government spending to help people pay for health insurance when people cannot pay for it all themselves?

1. Increase  2. Decrease  3. No change

*Reverse-coded:* Not applicable

**Gun regulation V202337**

*Original*: Do you think the federal government should make it more difficult for people to buy a gun than it is now, make it easier for people to buy a gun, or keep these rules about the same as they are now?

1. More difficult  2. Easier  3. Keep these rules about the same

*Reformulated:* What is this respondent's position on whether the federal government should make it more difficult for people to buy a gun than it is now, make it easier for people to buy a gun, or keep these rules about the same as they are now?

1. More difficult  2. Easier  3. Keep these rules about the same

*Reverse-coded:* Not applicable

**Income inequality V202257**

*Original*: Do you favor, oppose, or neither favor nor oppose the government trying to reduce the difference in incomes between the richest and poorest households?

1. Favor  2. Oppose  3. Neither favor nor oppose

*Reformulated:* How would this respondent assess whether the government should be trying to reduce the difference in incomes between the richest and poorest households?

1. Should be trying  2. Should not be trying  3. Neither of these

*Reverse-coded:* Do you favor, oppose, or neither favor nor oppose the government to stop trying to reduce the difference in incomes between the richest and poorest households?

1. Should be trying  2. Should not be trying  3. Neither of these

### A.9   Selected demographic variables

Table 24: Selected demographic variables of ANES 2020 respondents. 'Code' indicates the variable names used in the ANES 2020 dataset.

| Variable | Code | Answer choices |
|---|---|---|
| Race | V201549x | 1: white, 2: black, 3: hispanic, 4: asian, 5: native american |
| Gender | V201600 | 1: man, 2: woman |
| Age | V201507x | \<Open-ended\> |
| Ideology | V201200 | 1: extremely liberal, 2: liberal, 3: slightly liberal, 4: moderate, 5: slightly conservative, 6: conservative, 7: extremely conservative |
| Party | V201231x | 1: a strong democrat, 2: a weak Democrat, 3: an independent who leans Democratic, 4: an independent, 5: an independent who leans Republican, 6: a weak Republican, 7: a strong Republican |
| Political interest | V202406 | 1: very, 2: somewhat, 3: not very, 4: not at all |
| Church attendance | V201452 | 1: attend church, 2: do not attend church |
| Political discuss | V202022 | 1: I like to discuss politics with my family and friends., 2: I never discuss politics with my family or friends. |

### A.10   Example of prompts

We provide two prompt examples of GPT-4.1-mini simulations for user responses to the multiple-choice health insurance question, utilizing randomly sampled demographic variables. These examples illustrate a comparison under the Replicate and Reformulated question conditions.

---

**Replicate Condition**

**System**
Today is November 3, 2020. Racially, I am white. I like to discuss politics with my family and friends. Ideologically, I am moderate. Politically, I am a strong Republican. I attend church. I am 36 years old. I am a man.

**User**
Question: *Do you favor* an increase, decrease, or no change in government spending to help people pay for health insurance when people can't pay for it all themselves? Answer choices: 1. Increase 2. Decrease 3. No change When answering, respond ONLY with a single number that corresponds to the option you choose. Do not include any additional text, punctuation or explanation. My answer is

**Output**
1

---

---

**Reformulated Condition**

**System**
Today is November 3, 2020. Racially, *the respondent* is white. The respondent likes to discuss politics with their family and friends. Ideologically, the respondent is slightly liberal. Politically, the respondent is an independent who leans Democratic. The respondent does not attend church. The respondent is 37 years old. The respondent is a woman. The respondent is somewhat interested in politics.

**User**
Question: *How would this respondent assess* if there should be an increase, decrease, or no change in government spending to help people pay for health insurance when people cannot pay for it all themselves? Answer choices: 1. Increase 2. Decrease 3. No change When answering, respond ONLY with a single number that corresponds to the option you choose. Do not include any additional text, punctuation or explanation. My answer is

**Output**
1

---

