# OpenReview forum: "Mitigating Social Desirability Bias in Random Silicon Sampling"
_TMLR — Rejected by TMLR_

### Review · Reviewer_svrD · 2026-02-07

**Summary Of Contributions:**

This paper investigates whether any of four proposed prompting strategies are effective in reducing the divergence between the statistical distributions of responses to questions from the American National Election Study (ANES) derived from human participants and those produced by answers generated with LLMs. The latter of which is referred to as "silicon sampling." Divergence of such distributions has been documented in previous work and is especially present when questions consider sensitive topics or groups. Experiments in this work demonstrate that LLM prompts designed to include less evaluative language, third-personal formulations, and, when possible, specifically consider policies rather than groups or social phenomenon can produce better response alignment.

**Strengths**:

* The authors have performed a number of experiments across several models. One of the prompt techniques does show promise in reducing the Jensen-Shannon Divergence between model and human response distributions across the questions considered.

* The language of the paper is fairly clear and the authors include a lot of detail to help reproduce their experiments. I also appreciated the breakdown of results across demographic groups in the appendix.

* I think the visualizations presented in this paper are nicely organized and present a clear perspective on the impacts of the various approaches.

**Weaknesses**:

* The paper only considers models that have undergone alignment/human preference tuning. As the authors recognize, this likely heavily impacts the way such models respond to sensive questions or topics. Models like GPT-4.1 are also subject to an extensive number of hard (response filtration) and soft guardrails (system prompts). One of my biggest questions is whether models that do not undergo such a post-training phase demonstrate the same kinds of divergence? If not, then it seems as though the heuristic corrections propsed here are not necessary. Rather, silicon sampling should simply be done on model checkpoints before such mechanisms are applied.

* Jensen-Shannon Divergence (JSD) is the only measure of response distribution divergence. I am not an expert on statistical measures of survey distributions. However, there must be a deeply established practice in this field, which I suspect goes beyond JSD. For example, KL-Divergence is used in (Sun et al. 2024) in conjunction with Chi-squared tests. At the very least, the authors should argue why JSD is the right measure and why others need not be used.

* The use of third-person framing for prompts as a means of, at least partially, circumventing social desirability bias is a useful mechanism. However, I am concerned that such an approach risks encouraging models to invoke stereotyping as a means of estimating how someone with a narrowly defined profile might respond. There is evidence that profile-based generation does exactly this [[1]](https://aclanthology.org/2023.acl-long.84.pdf). The potential reliance on stereotypes to reflect true population level statistics is a notable risk.

* The results are based on a single study (ANES) across two different years. This makes it hard to know whether the proposed approach would generalize to other settings. Relatedly, I am unsure of the value of the results in Section 4.4 on temporal and demographic shift robustness. For the statistics in the appendix, there does not appear to be a significant shift in demographics between the ANES 2020 and 2024 cohorts. In the greedy decoding regime, if the prompts remain fixed, it's also unclear what would cause an LLM to change its estimate of demographic responses. There is a slight clue in the appendix that suggests the system prompts are dated, but it is not discussed in the main body. I suppose it is useful to know the approach does not collapse in this setting, but I'm not sure it is a meaningful test of robustness.

**Additional Comments:**

**Clarifications and Minor Comments**

1) The SDB acronym is defined multiple times in the body of the text and need only be when it first appears.

2) The alignment of the x-axis labels in Figure 2 (and others) feels a bit off. They are "centered" on the tick, which is atypical in my experience.

3) In Section 4.3, "Reformulated condition consistently reduces JS-divergence" does not seem grammatically correct.

4) Appendix A.4: Non-response is not formally defined. Does this mean that participants did not fill out an answer to the particular question?

5) Appendix A.7: Typo "Note for each tables questions..."

6) Somewhere in the body it should be explicitly stated that the prompts are zero-shot.

**Audience:**

Yes

**Audience Explanation:**

I do think some readers will be interested in the results of this work. However, I'm not sure if those with familiarity with prior work on this topic will draw significant new conclusions from the results.

**Broader Impact Concerns:**

I have no broader impact concerns with this work.

**Claims And Evidence:**

No

**Claims Explanation:**

I think there are some interesting elements to this work, but the effect of the reformulation strategy remains somewhat limited. In most instances the distributions remain fairly far apart and likely would not provide meaningful insight into real population dynamics. It's also unclear whether using post-alignment LLMs is even sensible for this kind of estimation. There are also a number aspects that I think need to be addressed or discussed further. See weaknesses and the comments below.

**Requested Changes:**

**Comments**

1) The citep and citet type citations are used incorrectly throughout and should be fixed.

2) The discussion of confounding factors that might affect LLM responses and their ability to be used for siicon sampling is, in my opinion not deep enough. In Sun et al. (2024), it is conjectured that social desirability bias is an underlying cause of the observed distribution divergences. However, to my knowledge, this is not confirmed. Thus, more discussion as to other aspects that might also contribute is warranted. Some existing work [[2]](https://pubsonline.informs.org/doi/10.1287/msom.2023.0279) has shown that LLMs demonstrate a mix of human- and non-human-like biases. Other work shows that LLMs tend toward response homogenization [[3]](https://openreview.net/forum?id=saDOrrnNTz), even under stochastic sampling, which could affect their ability to model response diversity. It seems worthwhile to at least discuss how other types of response bias might impact silicon sampling.

3) Based on some of the work cited in the paper, there seems to be at least some pre-existing work on reducing social desirability bias in human responses. Are there any existing corrections, beyond just question design, that could simply be applied to LLM responses?

4) Some statistics on how many respondents end up with the exact same demographic profile would be useful. In the greedy decoding regime, duplicate profiles means that the experiments are essentially querying the same individual twice.

5) The line "In the LLM setting, it may conversely trigger a perception of evaluation and increase conformity" struck me as odd. This is discussed in more detail in the actual results. However, it's placement here, without a discussion of why or at least a reference to the results, confused me as to why an LLM would have such a different response compared to humans.

6) It is, in some ways, unsurprising to me that the "Priming" approach was unsuccessful. It adds a significant twist to the demographic profiles.

7) There is no formal definition of a "socially acceptable" answer. If a definition from a prior study is being used, it should be cited. Otherwise, it is important to define this term. Relatedly, the vocabulary around "socially acceptable" answers should be tightened. For example, in at least one passage, these are referred to as "safe" (...more on the safe option in yellow...) answers which is not necessarily the same thing. Finally, the color scheme of Figure 2 (and others) is never clearly defined and only implicitly introduced in discussions of results. It appears the spectrum of blue to red indicates "socially acceptable" to "socially unacceptable" answers. Further, the methodology in how the answers were ranked with respect to social acceptability is not referred to in any place that I could find.

8) In presenting the response choices to the LLMs, were the answer options presented in the same order each time? LLMs can exhibit position bias when answering multiple-choice questions [[4]](https://openreview.net/forum?id=shr9PXz7T0).

9) There is no discussion of different regions having different socially desirable answers. There is an implicit assumption that LLMs inherit an American or North American bias, but it is never discussed.

10) In Section 4.3, it is unclear what is meant by "...alignment across seven aggregated demographic groups..." My best guess is the authors computed JSD for subgroups of Race (for example), computed the changes with different strategies, and then reported (in Table 2) the average changes, but that is not clear.

---

> ### Author Response · Authors · 2026-03-11
> **Response to Reviewer svrD (part 1/3)**
>
> We thank the reviewer for the valuable suggestions and constructive feedback. We have responded to the comments accordingly below. Looking forward to your response and further discussion.
>
>
> >**W1:** Performance of models that have not undergone alignment/human preference tuning.
>
>
> Thank you for the insightful comment. We agree that alignment and preference tuning likely influence how LLMs respond to socially sensitive questions and may contribute to social desirability bias (SDB). Our goal is to study SDB in silicon sampling applications, where researchers and practitioners typically rely on aligned, instruction-tuned, or API-access models rather than raw pretraining checkpoints. As an example of such applications, consider the marketing research field where most practitioners are not technically-trained and thus have to rely on API-access models for silicon sampling.
>
> Silicon sampling aims to approximate real-world survey deployment using LLM-based agents, which are designed to be safe, helpful, and norm-aware. From this perspective, SDB cannot be avoided by using base models, but reflects a characteristic of deployed systems that should be evaluated and addressed. In addition, base models often present practical challenges for survey simulation, such as weaker instruction following and unstable output formatting, and large-scale base checkpoints are often unavailable.
>
>
> We will clarify that our conclusions focus on aligned models and highlight studying SDB across the pretraining-alignment pipeline as a valuable direction for future work.
>
>
> >**W2:** Choice of JSD as the primary metric.
>
>
> Thank you for the comment. We have indeed considered KL-divergence first, but we chose JSD-divergence because it provides a more numerically stable, bounded, and symmetric measure, as discussed in the first paragraph of Section 3.4.  These properties are particularly important for comparing the distributions in our study. We will include a more detailed explanation in the revised paper.
>
>
> >**W3:** Potential for third-person prompting to induce stereotyping rather than reflecting true statistical populations when estimating demographic responses.
>
>
> Thank you for the insightful comment. We agree that profile-based prompting can potentially encourage models to rely on stereotypical associations when generating responses for demographic groups. In our study, the goal is to approximate aggregate response distributions for survey populations rather than to characterize or evaluate individuals. This inevitably requires the model to draw on associations it has learned about demographic groups, which may include both observed response patterns and undesirable stereotypes. We therefore treat this as a methodological limitation and discuss the ethical risks in the “Ethical Considerations” section. Our study does not assume that these generated demographic-response patterns are correct or socially justified. Instead, it evaluates whether prompt design can reduce social desirability bias when models are used as synthetic survey respondents.
>
>
> >**W4:** Limited generalizability due to results on a single dataset (ANES).
>
>
> Thank you for raising this concern. The purpose of Section 4.4 is not to claim broad robustness across different survey instruments or domains. Our goal is to examine whether the proposed reformulation strategy remains stable under realistic temporal variation within the same survey framework. To this end, the survey year is explicitly included in the prompt to provide temporal context (e.g., 2020 vs. 2024), as illustrated in Appendix A.10 (“Example of Prompts”). We will highlight this more clearly in the main text in the revision.
> While the demographic marginals between ANES 2020 and 2024 do not change dramatically, the political context and public opinion environment do evolve. Section 4.4 thus evaluates whether our method remains stable under such temporal shifts. A more detailed discussion of these changes is provided in Appendix A.4.
>
>
> We will clarify the purpose and moderate the wording to avoid overstating the scope. Evaluating the approach on additional surveys and domains is an important future direction, which we highlighted in the “Limitations” section.

---

> ### Author Response · Authors · 2026-03-11
> **Response to Reviewer svrD (part 2/3)**
>
> >**C1:** Incorrect usage of citation formats.
>
>
> Thank you for the comment, and we have fixed the issues in the new version.
>
>
> >**C2:** Need for broader discussion on how other types of response bias might impact silicon sampling.
>
>
> Thank you for the suggestion. We agree that SDB is unlikely to be the sole driver of distributional divergences between human and silicon responses, and that other factors may also contribute. Our study focuses on SDB as a measurable and theoretically grounded mechanism. In the revision, we will expand the discussion to acknowledge additional sources of bias documented in prior work and their potential implications for silicon sampling.
>
>
> >**C3:** Are there any existing corrections, beyond question design, that could be applied to LLM responses?
>
>
> Thank you for the insightful comment. In human survey methodology, responses are often adjusted post hoc via techniques such as calibration weighting [R1] and post-stratification [R2], which rely on external benchmark data (e.g., Census data) to correct sample biases. These approaches require validated ground-truth population information. In our setting, we treat LLMs as unsupervised survey respondents without access to such calibration signals or auxiliary validation data. In this context, survey design, such as question formulation and priming, is the primary mechanism available for influencing responses. Our study therefore focuses on prompt-level interventions, which we evaluate systematically. We will clarify this in the revised manuscript.
>
>
> [R1] Deville, Jean-Claude, and Carl-Erik Särndal. "Calibration estimators in survey sampling." Journal of the American statistical Association 87.418 (1992): 376-382.
>
>
> [R2] Royal, Kenneth D. "Survey research methods: A guide for creating post-stratification weights to correct for sample bias." Education in the Health Professions 2.1 (2019): 48-50.
>
>
> >**C4:** Statistics on duplicate demographic profiles would be useful.
>
>
> Thank you for the insightful comment. We analyzed demographic profile duplication when sampling 5,441 synthetic respondents (matching the size of ANES 2024) using GPT-4.1-mini under both Replicate and Reformulated conditions. The table below reports the percentage of duplicated demographic profiles across survey questions.
>
>
> |  | Current Economy | Gay Marriage | Refugee Allowing | Income Inequality | Gender Role | Climate Change | Gun Regulation | Drug Addiction | Race Diversity | Health Insurance | **Average** |
> |---|---|---|---|---|---|---|---|---|---|---|---|
> | **Replicate** | 2.55 | 2.30 | 2.57 | 2.57 | 2.33 | 2.26 | 2.41 | 2.22 | 2.33 | 2.72 | **2.43** |
> | **Reformulated** | 2.54 | 2.78 | 2.46 | 2.44 | 3.18 | 2.61 | 2.83 | 2.87 | 2.68 | 2.50 | **2.69** |
>
>
> Overall, duplication rates are low, averaging 2.43% under Replicate and 2.69% under Reformulated. Similarly low rates were observed in other conditions. This indicates that the vast majority of sampled profiles are unique, and duplicate profiles are unlikely to meaningfully affect results, even under greedy decoding. We will include this analysis and discussion in the appendix.
>
>
> >**C5:** Placing the statement "In the LLM setting, it may conversely trigger a perception of evaluation and increase conformity" in more context.
>
>
> Thank you for the suggestion. We agree that the statement may be confusing in this context without sufficient explanation. To avoid potential confusion, we will remove this sentence in the revised manuscript, as the detailed discussion already appears in the “Results” section.
>
>
> >**C6:** The "Priming" approach was unsurprisingly unsuccessful.
>
>
> We appreciate your observation. The priming condition was included to empirically evaluate this common strategy and illustrate its limitations.

---

> ### Author Response · Authors · 2026-03-11
> **Response to Reviewer svrD (part 3/3)**
>
> >**C7**: Definition of a socially acceptable answer.
>
>
> Thank you for the helpful suggestion. We will clarify the terminology and tighten the language in the revised manuscript. We provide a brief explanation below:
> In our study, we categorize answer options into three types. A **socially acceptable** (or desirable) response is one that expresses support for policies or positions generally framed as socially inclusive or supportive in the survey context. A **safe** (or moderate) response refers to a neutral option that avoids strong judgment or policy change, typically reflecting a status quo or middle position. A **socially undesirable** response refers to a more restrictive or opposing position relative to the policy or issue presented.
>
>
> For example, for the Health insurance question:
> “Do you favor an increase, decrease, or no change in government spending to help people pay for health insurance when people cannot pay for it all themselves?  1. Increase 2. Decrease 3. No change”
> In this case, *Increase* corresponds to the socially acceptable option, *Decrease* to the socially undesirable option, and *No change* to the safe option.
>
>
> We will also clarify the color scheme used in Figure 2 and related figures. Each color represents a distinct answer option, following the original ordering of response options in the survey. The color gradient itself does not encode social acceptability. This will be explicitly stated in the revised figure captions and text to avoid confusion.
>
>
> >**C8:** Potential for position bias in multiple-choice responses due to different answer ordering.
>
>
> Thank you for the insightful comment. In our experiments, answer options were presented in a fixed and consistent order across all conditions and models. Because our analyses are comparative (e.g., Reformulated vs. Replicate under identical option ordering), any potential position bias would affect all conditions equally and is therefore unlikely to confound our main conclusions. We will clarify this explicitly in the revised manuscript.
>
>
> >**C9:** No discussion of different regions.
>
>
> Thank you for the insightful comment. We agree that social desirability is highly culture-specific and that LLMs often reflect regional biases, such as Western or North American-centric perspectives. While a full cross-regional analysis is beyond the current scope of this study, we will expand our discussion to acknowledge this limitation and highlight regional comparative analysis as a critical direction for future work.
>
>
> >**C10:** The JSD calculations for demographic subgroups in Section 4.3.
>
>
> Thank you for your comment. Your interpretation is indeed correct: we computed the JSD for subgroups within each category (e.g., Race), calculated the changes under different strategies, and reported the average of these changes in Table 2. We will explicitly detail this calculation in the revised manuscript.
>
>
> >**C11:** Clarifications and minor comments on writing and formatting.
>
>
> Thank you for your helpful comments and we have addressed these issues in our revised manuscript.

---

> ### Comment · Reviewer_svrD · 2026-03-17
>
> **W1**: Thank you for this discussion. I think including the points you discuss here is a good way to motivate why studying techniques to mitigate SDB in the context of post-alignment models. While I still wonder whether pre-alignment models might side-step this issue, it is reasonable to leave it to future work.
>
> **W2**: The properties of JSD are useful in the context presented. My main concern is whether it is the "right" way to measure differences in survey distributions. It would be surprising to me if there is no standard statistical framework for measuring response distribution differences given the importance of survey responses and their long history of use. If JSD is one of the established methods, then a discussion and reference is certainly sufficient. As previously stated, even previous work uses a form of Chi-squared test in conjunction with divergence measures.
>
> **W3**: There are some nice points in the discussion you have provided here. I would recommend including them somewhere in the body and specifically discuss the risks associated with third-person prompting near where the technique is first introduced. This would include reference to [Marked Personas](https://aclanthology.org/2023.acl-long.84.pdf) and other work on the topic.
>
> **W4**: I do think the use of the 2024 ANES survey is useful, especially given the additional details that have been provided in the response. However, the primary component of my concern, which is the generalizability of the results to other response data, remains. As noted in the response, the 2024 survey is not meant to provide evidence of domain or survey robustness. From the current experimental results, it is difficult to infer whether the reformulation strategy will work for other settings.
>
> **C1-C6**: Thank you for these responses. They were helpful, and the proposed additions to the manuscript will be useful.
>
> **C7**: The provided definitions and clarifications are very helpful. I think the only additional piece that would be helpful is a sentence stating whether the categorization of responses was done by the authors themselves or if they exist as part of the original survey design.
>
> **C8**: The comparative nature of the study does ensure that when measuring differences between approaches, the effect would be present in all settings. However, it may cloud the overall divergence measurements. For example, in an extreme hypothetical case, if the difference between human and LLM response was strictly due to position bias, differences in responses due to prompt design would essentially be measuring how "well" the prompts combat position bias rather than SDB. I do not think position bias constitutes such a strong effect, and I am okay with the current design, but the proposed discussion of the experimental design and reference to the possibility of position bias would be useful.
>
> **C9-C11**: The proposed edits and additional discussion are appreciated.

---

### Review · Reviewer_k9W3 · 2026-03-03

**Summary Of Contributions:**

This paper explores the impact of Social Desirability Bias (SDB) in "silicon sampling," the use of LLMs to simulate survey responses from entire populations. The authors conduct a systematic study that tests four prompting strategies (third-person/neutral reformulation, reverse-coding of questions, analytic priming, and sincerity preamble) across three models (Llama-3.1-8B, Llama-3.1-70B, and GPT-4.1-mini). Using data from the ANES as a human benchmark, the study demonstrates that reformulating neutral prompts is the most effective technique for mitigating the tendency to focus on socially acceptable options, significantly improving the alignment between the synthetic and real human distributions. In contrast, the other strategies yield unstable results or even increase response uniformity.

**Audience:**

Yes

**Audience Explanation:**

The practice of using LLM agents to simulate human participants in psychological studies, surveys, and market research is attracting considerable attention in the ML and social science communities. Comprehending where these models separate from real human data, particularly on socially sensitive topics, and how prompt engineering can correct or fail to correct these biases is quite useful information for anyone working on model alignment and synthetic data validity.

**Broader Impact Concerns:**

The paper is in fine condition. However, I believe this assessment requires some additions to be fully comprehensive.

First, the authors should more incisively discuss the long-term risks associated with these mitigation techniques. Only superficially mitigating "Social Desirability Bias" risks instilling a false sense of security in researchers: they could be led to believe they have generated a perfectly representative sample, when in reality the model may lack internal representations and real knowledge of the true opinions of specific demographic subgroups. In these scenarios where there is a genuine knowledge limitation, simply removing superficial bias could prove not only ineffective but even misleading.

Second, it is crucial to emphasise clearly that the results obtained are based on ANES data and are therefore strictly limited to the US demographic and sociopolitical context. Authors should explicitly caution readers that the effectiveness of such strategies should not be assumed globally or applied to other cultural contexts and datasets without first conducting further rigorous empirical testing.

**Claims And Evidence:**

Yes

**Claims Explanation:**

The authors' claims are supported by a rigorous methodological analysis. Specifically, the Jensen-Shannon divergence is used to mathematically quantify the alignment between the ANES human data and those generated by the LLMs.

To ensure the statistical reliability of the results and to quantify uncertainty, the authors use bootstrap sampling to construct 95% confidence intervals.

Finally, the study verifies the robustness of its conclusions by conducting cross-sectional tests across demographic categories and comparing survey waves (ANES 2020 vs ANES 2024).

**Requested Changes:**

While the paper is solid, I recommend a few changes to improve contextualization and analysis:

I recommend including related works on de-biasing in large language models, such as "A Trip Towards Fairness: Bias and De-biasing in Large Language Models," in the Related Work section. Including this reference would greatly enrich the context on current bias measurement and mitigation techniques in LLMs, offering a broader perspective to your readers.

It would be helpful to briefly expand the discussion in the limitations section on why the reframing technique is less effective for economic topics than for those related to social norms and identity. In particular, I encourage you to explore the hypothesis that this effect intersects with the phenomenon of sycophancy. Indeed, models may show greater resistance to debiasing on economic topics because they tend to adopt a compliant attitude, attempting to align themselves with what they perceive as the user's expectations or opinions. In this regard, I strongly suggest citing and connecting to the dynamics discussed in the paper "When do large language models contradict humans? Large language models' sycophantic behaviour," which would provide an excellent theoretical lens to explain this discrepancy in your results.

---

> ### Author Response · Authors · 2026-04-01
> **Response to Reviewer k9W3**
>
> We thank the reviewer for these insightful and constructive suggestions.
>
> **Related work in de-biasing.** We agree that situating our work more broadly within the literature on bias and de-biasing in LLMs would strengthen the paper. In the revision, we will expand the Related Work section to include relevant studies and better contextualize our work within existing bias mitigation frameworks.
>
> **Effectiveness of reframing across topics.** We appreciate the suggestion to further analyze why reframing is less effective for economic topics compared to social or identity-related ones. We also thank the reviewer for highlighting the potential connection to sycophantic behavior and for pointing us to relevant work. In the revision, we will expand the discussion to explore this hypothesis and better explain the observed discrepancy across topic domains.
>
> **Long-term risks of mitigation.** We will more explicitly discuss the risk that superficial mitigation may create a false sense of reliability, especially when models lack grounded knowledge of subgroup-specific opinions, as noted in our limitations “Bias vs. knowledge limitations”. We will clarify these points in the ethical considerations.
>
> **For the scope and generalizability.** In the limitation section, we agree that our findings are grounded in American National Election Studies data and therefore reflect only a U.S.-specific sociopolitical context. We will make this limitation more explicit and caution against directly generalizing our conclusions to other cultural settings or datasets without further empirical validation.

---

### Review · Reviewer_SYn6 · 2026-03-26

**Summary Of Contributions:**

Language models have the aim of matching human survey distributions on questions for use in simulating survey responses. However, the primary finding in this work is a drift toward socially approved answers on sensitive topics. To close this gap, the authors test whether prompt framing reduces normative pressure and brings silicon samples closer to human baseline data. They find that simple, neutral third-person reformulations effectively improve alignment and reduce JS-divergence. Conversely, they test and observe that fancier techniques like priming and reverse-coding fail, often making things worse by increasing response uniformity. Demographic-stratified analyses further show that reformulation improves alignment on many topics.

I find this work timely as there would be an increasing use of language models for simulating surveys. Therefore, understanding model drift on sensitive items is critical, and try to provide practical prompt changes which improve realism.

**Audience:**

Yes

**Audience Explanation:**

Yes. I think the paper would interest readers working on LLM evaluation in synthetic populations, could be very useful for computational social science. The work is timely, it provides valueable empirical signal for a bias and simple way of improving it -- very practical in my opinion.

**Claims And Evidence:**

Yes

**Claims Explanation:**

Parts which are supported (pros):

- There are several methodological strengths. The authors conduct extensive statistical analyses, comparing three models while including bootstrapped uncertainty, vary the temperature, subgroup evaluations (especially liked this one), and temporal testing. Furthermore, the I liked the detailed analysis across subgroups in Appendix. It supports both the core issue raised and using third-person prompting addressing the core issue.

Parts which less-supported (cons):

- I worry the evidence is narrower than the authors claim (from both models and datapoints angle).
    - Model: The observations are clean (over-concentration on socially acceptable answers) on sensitive items only in Llama 8B. I did not see large effects on other models. This suggests the issue may be a issue in Llama 8B rather than a general failure across "silicon sampling" i.e. language models.
    - Data: The main study relies on ten questions from a single U.S. survey, and the temporal robustness check uses only eight. Because the models were released after the survey dates,I'm not sure how well it tests out-of-time generalization.

- How reflective is the sampling methodology w.r.t real world effect?
    - The work samples demographic variables *independently* from marginal distributions. Won't it decorrelate real-world correlations among race, age, party, ideology, and religion -- I wouldn't be surprised if it generates implausible respondent profiles which causes some of these issues. How reliable is the independent sampling assumption, to what degree might it cause the underlying problem or affect the realism of results?

Overall, I liked the work and think the evidence sufficiently supports the claims.

**Requested Changes:**

Could you please address the weaknesses?

---

> ### Author Response · Authors · 2026-04-01
> **Response to Reviewer SYn6 (part 1/2)**
>
> We thank the reviewer for the valuable feedback. We have responded to the comments accordingly below. Looking forward to your response and further discussion.
>
>
> >**W1:** The evidence might be narrower than the authors claim from the model angle. The observations are clean (over-concentration on socially acceptable answers) on sensitive items only in Llama 8B. No large effects on other models.
>
>
> We thank the reviewer for this important concern. We agree that the over-concentration on socially acceptable answers is most pronounced in Llama 8B. However, we also observe similar patterns in Llama 70B and GPT-4.1-mini, though on fewer questions and with smaller magnitudes.
>
>
> We acknowledge that this suggests the effect may be more prominent in smaller models rather than universal across all LLMs. In Section 4.1, we discussed that this behavior may be related to social desirability bias (SDB) being amplified in lower-capacity models. Larger models exhibit fewer instances of such over-concentration, which may reflect improved ability to represent more diverse response distributions.
>
>
> We will clarify this distinction in the revised manuscript and position Llama 8B as a stronger (but not exclusive) case of the phenomenon.
>
>
> >**W2:** The evidence might be narrower than the authors claim from the datapoint angle. The main study relies on ten questions, and the temporal robustness check uses only eight.
>
>
> We thank the reviewer for this important observation.
>
>
> **For the number of questions.** We follow Sun et al. (2024) in selecting the ten survey questions to enable a standardized comparison with prior work, and these questions span diverse societal topics and dimensions.
>
>
> For the temporal robustness analysis, we restrict to eight overlapping questions that are directly comparable between ANES 2020 and 2024, as the remaining two differ in semantics and answer choices. We consider such a restriction to ensure a fair comparison when evaluating the effect of our mitigation strategy (i.e., reformulation).
>
>
> We agree that the overall number of questions is limited and may constrain the breadth of our conclusions. We will explicitly acknowledge this as a limitation and clarify that expanding the question set is an important direction for future work.
>
>
> **Out-of-time generalization.** We acknowledge that some models were released after the survey years and may have been exposed to related information during training, which limits a strict interpretation of out-of-time generalization. However, our goal is to examine how LLM responses change under different prompt formulations of the same survey question (e.g., first-person vs. third-person framing). We isolate the SDB mitigation effect by holding question content fixed and varying only the prompt structure.
>
>
> Additionally, our approach focuses on generating responses at the level of individual profiles (i.e., silicon samples), rather than reproducing aggregate survey statistics. This reduces the likelihood that results are driven purely by memorization of population-level distributions, although we agree that some degree of information leakage cannot be fully ruled out.
>
>
> We will clarify this distinction in the paper and explicitly state the limitation. We acknowledge that evaluating true out-of-time generalization is an important direction for future work, e.g., using newer surveys conducted after model release.

---

> ### Author Response · Authors · 2026-04-01
> **Response to Reviewer SYn6 (part 2/2)**
>
> >**W3:** How reflective is the sampling methodology w.r.t real world effect? Effect of implausible respondent profiles.
>
>
> We thank the reviewer for raising this important concern. We agree that independently sampling demographic attributes from their empirical marginal distributions does not preserve real-world correlations (e.g., between age, party, and ideology), and may result in implausible combinations.
>
>
> Our goal in using independent sampling (“random silicon sample”) is to ensure broad and systematic coverage of the demographic space, allowing us to probe how models respond across a wide range of profiles in a controlled manner. This design helps probe the effect of individual attributes and their interactions with prompt formulation, rather than entangling them with a fixed empirical distribution. Moreover, the prior study (Sun et al. 2024) already tested how the model responses differ between random silicon samples and a fully replicated ANES sample (“silicon sample”) for presidential voting (see Table 1 in Sun et al. (2024)). They found that through random silicon sampling, the language model generated a response distribution that is remarkably similar to the actual ANES responses (with lower KL-divergence), compared to the full replicate (“silicon sample”). Their observation also confirms the effectiveness of independent sampling.
>
>
> We acknowledge that not all sampled profiles are equally plausible, which may limit how directly the results reflect real-world populations. Using more realistic joint distributions could improve fidelity to real-world populations, but it also introduces its own **challenges**. For example, sampling from an empirical joint distribution (e.g., from ANES) may overrepresent dominant demographic combinations (e.g., older voters with consistent party and ideology), while under-sampling rare profiles. In addition, approaches such as “silicon sampling” requires demographic information about individuals within the group, which is challenging to obtain in actual survey scenarios (Sun et al. 2024).
>
>
> We will clarify this trade-off in the revised manuscript and explicitly highlight it as a limitation. As future work, we plan to explore more realistic alternatives, such as sampling from estimated joint distributions or using conditional generation methods that better preserve demographic correlations, and to compare these approaches against independent sampling in evaluating SDB mitigation effects.

---

### Decision · Action_Editor_UoJM · 2026-05-29

**Recommendation:** Reject

**Additional Comments:**

A revised manuscript should include the following:
* experiments on additional datasets or a convincing rationale why the presented experiments are sufficient to back up the claims
* the promised clarifications, including those with respect to the results’ generlizability, risks of superficial bias mitigation
* a detailed discussion of related work on de-biasing methods for LLMs

**Audience:**

Yes

**Audience Explanation:**

The work is a well-motivated study on an increasingly important topic. I am convinced that several members of the TMLR audience would be interested in the findings of this paper.

**Claims And Evidence:**

No

**Claims Explanation:**

This well-written manuscript demonstrates the presence of social desirability bias in LLM-based survey responses: LLMs show the tendency to generate socially acceptable answers when confronted with sensitive questions. To address this issue, the authors explore prompt-based strategies, among which prompt reformulation appeared to be the most effective.

Reviewers acknowledged the overall rigour of the statistical analysis, with the exception of Reviewer svrD, who posed the question whether Jensen-Shannon dIvergence is an established metric for comparing survey response distributions or if maybe additional metrics should have been reported for a more complete picture.

Reviewers also have noted the limitation of the study only considering two waves of the ANES survey, which limits my confidence in the general nature of claims.
During the rebuttal, authors mentioned that they were going to make several updates to the manuscript, including toning down some of the claims. However, there is no new revision since January 27 (before the reviewing phase).

In the current state the manuscript does not seem to be ready for publication. See the expected changes below.

**Resubmission Of Major Revision:**

The authors may consider submitting a major revision at a later time.